# RETHINKING ATTENTION WITH PERFORMERS

**Krzysztof Choromanski**[*1], **Valerii Likhosherstov**[*2], **David Dohan**[*1], **Xingyou Song**[*1]
**Andreea Gane**[*1], **Tamas Sarlos**[*1], **Peter Hawkins**[*1], **Jared Davis**[*3], **Afroz Mohiuddin**[1]
**Lukasz Kaiser**[1], **David Belanger**[1], **Lucy Colwell**[1,2], **Adrian Weller**[2,4]
[1]Google [2]University of Cambridge [3]DeepMind [4]Alan Turing Institute

## ABSTRACT

We introduce *Performers*, Transformer architectures which can estimate regular (softmax) full-rank-attention Transformers with provable accuracy, but using only linear (as opposed to quadratic) space and time complexity, without relying on any priors such as sparsity or low-rankness. To approximate softmax attention-kernels, Performers use a novel *Fast Attention Via positive Orthogonal Random features* approach (FAVOR+), which may be of independent interest for scalable kernel methods. FAVOR+ can also be used to efficiently model kernelizable attention mechanisms beyond softmax. This representational power is crucial to accurately compare softmax with other kernels for the first time on large-scale tasks, beyond the reach of regular Transformers, and investigate optimal attention-kernels. Performers are linear architectures fully compatible with regular Transformers and with strong theoretical guarantees: unbiased or nearly-unbiased estimation of the attention matrix, uniform convergence and low estimation variance. We tested Performers on a rich set of tasks stretching from pixel-prediction through text models to protein sequence modeling. We demonstrate competitive results with other examined efficient sparse and dense attention methods, showcasing effectiveness of the novel attention-learning paradigm leveraged by Performers.

## 1 INTRODUCTION AND RELATED WORK

Transformers (Vaswani et al., 2017; Dehghani et al., 2019) are powerful neural network architectures that have become SOTA in several areas of machine learning including natural language processing (NLP) (e.g. speech recognition (Luo et al., 2020)), neural machine translation (NMT) (Chen et al., 2018), document generation/summarization, time series prediction, generative modeling (e.g. image generation (Parmar et al., 2018)), music generation (Huang et al., 2019), and bioinformatics (Rives et al., 2019; Madani et al., 2020; Ingraham et al., 2019; Elnaggar et al., 2019; Du et al., 2020).

Transformers rely on a trainable *attention* mechanism that identifies complex dependencies between the elements of each input sequence. Unfortunately, the regular Transformer scales quadratically with the number of tokens $L$ in the input sequence, which is prohibitively expensive for large $L$ and precludes its usage in settings with limited computational resources even for moderate values of $L$. Several solutions have been proposed to address this issue (Beltagy et al., 2020; Gulati et al., 2020; Chan et al., 2020; Child et al., 2019; Bello et al., 2019). Most approaches restrict the attention mechanism to attend to local neighborhoods (Parmar et al., 2018) or incorporate structural priors on attention such as sparsity (Child et al., 2019), pooling-based compression (Rae et al., 2020) clustering/binning/convolution techniques (e.g. (Roy et al., 2020) which applies $k$-means clustering to learn dynamic sparse attention regions, or (Kitaev et al., 2020), where locality sensitive hashing is used to group together tokens of similar embeddings), sliding windows (Beltagy et al., 2020), or truncated targeting (Chelba et al., 2020). There is also a long line of research on using dense attention matrices, but defined by low-rank kernels substituting softmax (Katharopoulos et al., 2020; Shen et al., 2018). Those methods critically rely on kernels admitting explicit representations as dot-products of finite positive-feature vectors.

The approaches above do not aim to approximate regular attention, but rather propose simpler and more tractable attention mechanisms, often by incorporating additional constraints (e.g. identical query and key sets as in (Kitaev et al., 2020)), or by trading regular with sparse attention using more

---

[*]Equal contribution. Correspondence to {kchoro,lcolwell}@google.com.

Code for Transformer models on protein data can be found in `github.com/google-research/google-research/tree/master/protein_lm` and Performer code can be found in `github.com/google-research/google-research/tree/master/performer`. Google AI Blog: `https://ai.googleblog.com/2020/10/rethinking-attention-with-performers.html`

layers (Child et al., 2019). Unfortunately, there is a lack of rigorous guarantees for the representation power produced by such methods, and sometimes the validity of sparsity patterns can only be verified empirically through trial and error by constructing special GPU operations (e.g. either writing C++ CUDA kernels (Child et al., 2019) or using TVMs (Beltagy et al., 2020)). Other techniques which aim to reduce Transformers' space complexity include reversible residual layers allowing one-time activation storage in training (Kitaev et al., 2020) and shared attention weights (Xiao et al., 2019). These constraints may impede application to long-sequence problems, where approximations of the attention mechanism are not sufficient. Approximations based on truncated back-propagation (Dai et al., 2019) are also unable to capture long-distance correlations since the gradients are only propagated inside a localized window. Other methods propose biased estimation of regular attention but only in the non-causal setting and with large mean squared error (Wang et al., 2020).

In response, we introduce the first Transformer architectures, *Performers*, capable of **provably** accurate and practical estimation of regular (softmax) full-rank attention, but of only linear space and time complexity and **not relying on any priors** such as sparsity or low-rankness. Performers use the *Fast Attention Via positive Orthogonal Random features* (FAVOR+) mechanism, leveraging new methods for approximating softmax and Gaussian kernels, which we propose. We believe these methods are of independent interest, contributing to the theory of scalable kernel methods. Consequently, Performers are the first linear architectures **fully compatible** (via small amounts of fine-tuning) with regular Transformers, providing strong theoretical guarantees: unbiased or nearly-unbiased estimation of the attention matrix, uniform convergence and lower variance of the approximation.

FAVOR+ can be also applied to efficiently model other kernelizable attention mechanisms beyond softmax. This representational power is crucial to accurately compare softmax with other kernels for the first time on large-scale tasks, that are beyond the reach of regular Transformers, and find for them optimal attention-kernels. FAVOR+ can also be applied beyond the Transformer scope as a more scalable replacement for regular attention, which itself has a wide variety of uses in computer vision (Fu et al., 2019), reinforcement learning (Zambaldi et al., 2019), training with softmax cross entropy loss, and even combinatorial optimization (Vinyals et al., 2015).

We test Performers on a rich set of tasks ranging from pixel-prediction through text models to protein sequence modeling. We demonstrate competitive results with other examined efficient sparse and dense attention methods, showcasing the effectiveness of the novel attention-learning paradigm leveraged by Performers. We emphasize that in principle, FAVOR+ can also be combined with other techniques, such as reversible layers (Kitaev et al., 2020) or cluster-based attention (Roy et al., 2020).

## 2 FAVOR+ MECHANISM & POSITIVE ORTHOGONAL RANDOM FEATURES

Below we describe in detail the FAVOR+ mechanism - the backbone of the $\mathrm{Performer's}$ architecture. We introduce a new method for estimating softmax (and Gaussian) kernels with **positive** orthogonal random features which FAVOR+ leverages for the robust and unbiased estimation of regular (softmax) attention and show how FAVOR+ can be applied for other attention-kernels.

### 2.1 PRELIMINARIES - REGULAR ATTENTION MECHANISM

Let $L$ be the size of an input sequence of tokens. Then regular dot-product attention (Vaswani et al., 2017) is a mapping which accepts matrices $\mathbf{Q}, \mathbf{K}, \mathbf{V} \in \mathbb{R}^{L \times d}$ as input where $d$ is the hidden dimension (dimension of the latent representation). Matrices $\mathbf{Q}, \mathbf{K}, \mathbf{V}$ are intermediate representations of the input and their rows can be interpreted as *queries*, *keys* and *values* of the continuous dictionary data structure respectively. *Bidirectional (or non-directional (Devlin et al., 2018)) dot-product attention* has the following form, where $\mathbf{A} \in \mathbb{R}^{L \times L}$ is the so-called *attention matrix*:

$$\mathrm{Att}_{\leftrightarrow}(\mathbf{Q}, \mathbf{K}, \mathbf{V}) = \mathbf{D}^{-1} \mathbf{A} \mathbf{V}, \quad \mathbf{A} = \exp(\mathbf{Q} \mathbf{K}^{\top} / \sqrt{d}), \quad \mathbf{D} = \mathrm{diag}(\mathbf{A} \mathbf{1}_L). \quad (1)$$

Here $\exp(\cdot)$ is applied elementwise, $\mathbf{1}_L$ is the all-ones vector of length $L$, and $\mathrm{diag}(\cdot)$ is a diagonal matrix with the input vector as the diagonal. Time and space complexity of computing (1) are $O(L^2 d)$ and $O(L^2 + Ld)$ respectively, because $\mathbf{A}$ has to be stored explicitly. Hence, in principle, dot-product attention of type (1) is incompatible with end-to-end processing of long sequences. Bidirectional attention is applied in encoder self-attention and encoder-decoder attention in Seq2Seq architectures.

Another important type of attention is *unidirectional dot-product attention* which has the form:

$$\mathrm{Att}_{\rightarrow}(\mathbf{Q}, \mathbf{K}, \mathbf{V}) = \widetilde{\mathbf{D}}^{-1} \widetilde{\mathbf{A}} \mathbf{V}, \quad \widetilde{\mathbf{A}} = \mathrm{tril}(\mathbf{A}), \quad \widetilde{\mathbf{D}} = \mathrm{diag}(\widetilde{\mathbf{A}} \mathbf{1}_L), \quad (2)$$

where $\mathrm{tril}(\cdot)$ returns the lower-triangular part of the argument matrix including the diagonal. As discussed in (Vaswani et al., 2017), unidirectional attention is used for autoregressive generative modelling, e.g. as self-attention in generative Transformers as well as the decoder part of Seq2Seq Transformers.

We will show that attention matrix $\mathbf{A}$ can be approximated up to any precision in time $O(Ld^2 \log(d))$. For comparison, popular methods leveraging sparsity via Locality-Sensitive Hashing (LSH) techniques (Kitaev et al., 2020) have $O(Ld^2 \log L)$ time complexity. In the main body of the paper we will describe FAVOR+ for bidirectional attention. Completely analogous results can be obtained for the unidirectional variant via the mechanism of *prefix-sums* (all details in the Appendix B.1).

## 2.2 GENERALIZED KERNELIZABLE ATTENTION

FAVOR+ works for attention blocks using matrices $\mathbf{A} \in \mathbb{R}^{L \times L}$ of the form $\mathbf{A}(i, j) = \mathrm{K}(\mathbf{q}_i^\top, \mathbf{k}_j^\top)$, with $\mathbf{q}_i/\mathbf{k}_j$ standing for the $i^{th}/j^{th}$ query/key row-vector in $\mathbf{Q}/\mathbf{K}$ and kernel $\mathrm{K} : \mathbb{R}^d \times \mathbb{R}^d \to \mathbb{R}_+$ defined for the (usually randomized) mapping: $\phi : \mathbb{R}^d \to \mathbb{R}_+^r$ (for some $r > 0$) as:

$$\mathrm{K}(\mathbf{x}, \mathbf{y}) = \mathbb{E}[\phi(\mathbf{x})^\top \phi(\mathbf{y})]. \tag{3}$$

We call $\phi(\mathbf{u})$ a *random feature map* for $\mathbf{u} \in \mathbb{R}^d$. For $\mathbf{Q}', \mathbf{K}' \in \mathbb{R}^{L \times r}$ with rows given as $\phi(\mathbf{q}_i^\top)^\top$ and $\phi(\mathbf{k}_i^\top)^\top$ respectively, Equation 3 leads directly to the efficient attention mechanism of the form:

$$\widehat{\mathrm{Att}_\leftrightarrow}(\mathbf{Q}, \mathbf{K}, \mathbf{V}) = \widehat{\mathbf{D}}^{-1}(\mathbf{Q}'((\mathbf{K}')^\top \mathbf{V})), \qquad \widehat{\mathbf{D}} = \mathrm{diag}(\mathbf{Q}'((\mathbf{K}')^\top \mathbf{1}_L)). \tag{4}$$

Here $\widehat{\mathrm{Att}_\leftrightarrow}$ stands for the approximate attention and brackets indicate the order of computations. It is easy to see that such a mechanism is characterized by space complexity $O(Lr + Ld + rd)$ and time complexity $O(Lrd)$ as opposed to $O(L^2 + Ld)$ and $O(L^2d)$ of the regular attention (see also Fig. 1).

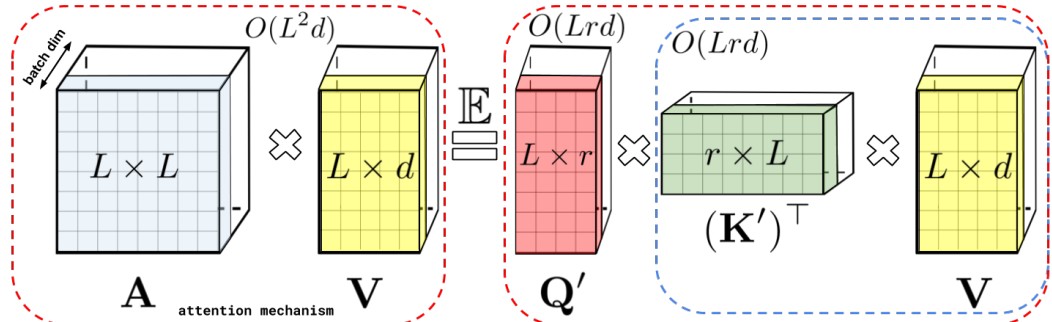

Figure 1: Approximation of the regular attention mechanism $\mathbf{AV}$ (before $\mathbf{D}^{-1}$-renormalization) via (random) feature maps. Dashed-blocks indicate order of computation with corresponding time complexities attached.

The above scheme constitutes the FA-part of the FAVOR+ mechanism. The remaining OR+ part answers the following questions: **(1)** How expressive is the attention model defined in Equation 3, and in particular, can we use it in principle to approximate regular softmax attention ? **(2)** How do we implement it robustly in practice, and in particular, can we choose $r \ll L$ for $L \gg d$ to obtain desired space and time complexity gains? We answer these questions in the next sections.

## 2.3 HOW TO AND HOW NOT TO APPROXIMATE SOFTMAX-KERNELS FOR ATTENTION

It turns out that by taking $\phi$ of the following form for functions $f_1, ..., f_l : \mathbb{R} \to \mathbb{R}$, function $g : \mathbb{R}^d \to \mathbb{R}$ and deterministic vectors $\omega_i$ or $\omega_1, ..., \omega_m \overset{\mathrm{iid}}{\sim} \mathcal{D}$ for some distribution $\mathcal{D} \in \mathcal{P}(\mathbb{R}^d)$:

$$\phi(\mathbf{x}) = \frac{h(\mathbf{x})}{\sqrt{m}}(f_1(\omega_1^\top \mathbf{x}), ..., f_1(\omega_m^\top \mathbf{x}), ..., f_l(\omega_1^\top \mathbf{x}), ..., f_l(\omega_m^\top \mathbf{x})), \tag{5}$$

we can model most kernels used in practice. Furthermore, in most cases $\mathcal{D}$ is isotropic (i.e. with pdf function constant on a sphere), usually Gaussian. For example, by taking $h(\mathbf{x}) = 1$, $l = 1$ and $\mathcal{D} = \mathcal{N}(0, \mathbf{I}_d)$ we obtain estimators of the so-called PNG-kernels (Choromanski et al., 2017) (e.g. $f_1 = \mathrm{sgn}$ corresponds to the angular kernel). Configurations: $h(\mathbf{x}) = 1$, $l = 2$, $f_1 = \sin$, $f_2 = \cos$ correspond to shift-invariant kernels, in particular $\mathcal{D} = \mathcal{N}(0, \mathbf{I}_d)$ leads to the Gaussian kernel $\mathrm{K}_{\mathrm{gauss}}$ (Rahimi & Recht, 2007). The *softmax-kernel* which defines regular attention matrix $\mathbf{A}$ is given as:

$$\mathrm{SM}(\mathbf{x}, \mathbf{y}) \stackrel{\text{def}}{=} \exp(\mathbf{x}^\top \mathbf{y}). \tag{6}$$

In the above, without loss of generality, we omit $\sqrt{d}$-renormalization since we can equivalently renormalize input keys and queries. Since: $\mathrm{SM}(\mathbf{x}, \mathbf{y}) = \exp(\frac{\|\mathbf{x}\|^2}{2}) \mathrm{K}_{\mathrm{gauss}}(\mathbf{x}, \mathbf{y}) \exp(\frac{\|\mathbf{y}\|^2}{2})$, based on what we have said, we obtain random feature map unbiased approximation of $\mathrm{SM}(\mathbf{x}, \mathbf{y})$ using trigonometric functions with: $h(\mathbf{x}) = \exp(\frac{\|\mathbf{x}\|^2}{2})$, $l = 2$, $f_1 = \sin$, $f_2 = \cos$. We call it $\widehat{\mathrm{SM}}_m^{\mathrm{trig}}(\mathbf{x}, \mathbf{y})$.

There is however a caveat there. The attention module from (1) constructs for each token, a convex combination of value-vectors with coefficients given as corresponding renormalized kernel scores. That is why kernels producing non-negative scores are used. Applying random feature maps with potentially negative dimension-values ($\sin / \cos$) leads to unstable behaviours, especially when kernel scores close to 0 (which is the case for many entries of $\mathbf{A}$ corresponding to low relevance tokens) are approximated by estimators with large variance in such regions. This results in abnormal behaviours, e.g. negative-diagonal-values renormalizers $\mathbf{D}^{-1}$, and consequently either completely prevents training or leads to sub-optimal models. We demonstrate empirically that this is what happens for $\widehat{\mathrm{SM}}_m^{\mathrm{trig}}$ and provide detailed theoretical explanations showing that the variance of $\widehat{\mathrm{SM}}_m^{\mathrm{trig}}$ is large as approximated values tend to 0 (see: Section 3). This is one of the main reasons why the robust random feature map mechanism for approximating regular softmax attention was never proposed.

We propose a robust mechanism in this paper. Furthermore, the variance of our new unbiased positive random feature map estimator tends to 0 as approximated values tend to 0 (see: Section 3).

**Lemma 1** (Positive Random Features (PRFs) for Softmax). *For $\mathbf{x}, \mathbf{y} \in \mathbb{R}^d$, $\mathbf{z} = \mathbf{x} + \mathbf{y}$ we have:*

$$\mathrm{SM}(\mathbf{x}, \mathbf{y}) = \mathbb{E}_{\omega \sim \mathcal{N}(0, \mathbf{I}_d)} \Big[ \exp\Big( \omega^\top \mathbf{x} - \frac{\|\mathbf{x}\|^2}{2} \Big) \exp\Big( \omega^\top \mathbf{y} - \frac{\|\mathbf{y}\|^2}{2} \Big) \Big] = \Lambda \mathbb{E}_{\omega \sim \mathcal{N}(0, \mathbf{I}_d)} \cosh(\omega^\top \mathbf{z}), \tag{7}$$

*where $\Lambda = \exp(-\frac{\|\mathbf{x}\|^2 + \|\mathbf{y}\|^2}{2})$ and $\cosh$ is hyperbolic cosine. Consequently, softmax-kernel admits a positive random feature map unbiased approximation with $h(\mathbf{x}) = \exp(-\frac{\|\mathbf{x}\|^2}{2})$, $l = 1$, $f_1 = \exp$ and $\mathcal{D} = \mathcal{N}(0, \mathbf{I}_d)$ or: $h(\mathbf{x}) = \frac{1}{\sqrt{2}} \exp(-\frac{\|\mathbf{x}\|^2}{2})$, $l = 2$, $f_1(u) = \exp(u)$, $f_2(u) = \exp(-u)$ and the same $\mathcal{D}$ (the latter for further variance reduction). We call related estimators: $\widehat{\mathrm{SM}}_m^+$ and $\widehat{\mathrm{SM}}_m^{\mathrm{hyp}+}$.*

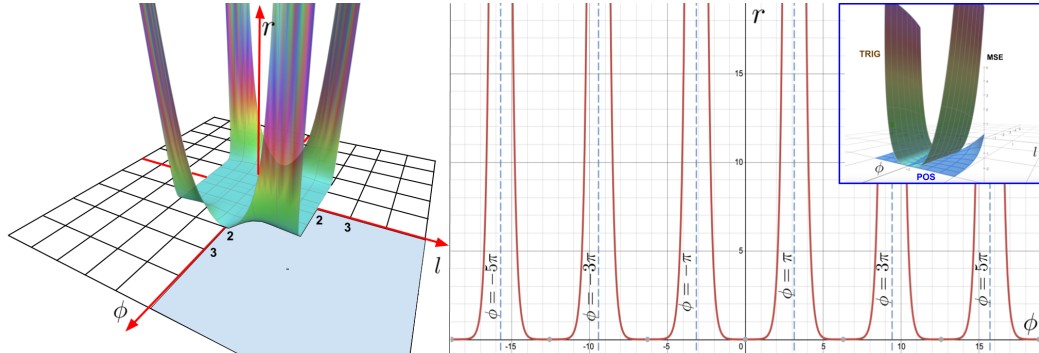

Figure 2: **Left:** Symmetrized (around origin) utility function $r$ (defined as the ratio of the mean squared errors (MSEs) of estimators built on: trigonometric and positive random features) as a function of the angle $\phi$ (in radians) between input feature vectors and their lengths $l$. Larger values indicate regions of $(\phi, l)$-space with better performance of positive random features. We see that for critical regions with $\phi$ large enough (small enough softmax-kernel values) our method is arbitrarily more accurate than trigonometric random features. Plot presented for domain $[-\pi, \pi] \times [-2, 2]$. **Right:** The slice of function $r$ for fixed $l = 1$ and varying angle $\phi$. **Right Upper Corner:** Comparison of the MSEs of both the estimators in a low softmax-kernel value region.

In Fig. 2 we visualize the advantages of positive versus standard trigonometric random features. In critical regions, where kernel values are small and need careful approximation, our method outperforms its counterpart. In Section 4 we further confirm our method's advantages empirically, using positive features to efficiently train softmax-based linear Transformers. If we replace in (7) $\omega$ with $\sqrt{d} \frac{\omega}{\|\omega\|}$, we obtain the so-called **regularized softmax-kernel** SMREG which we can approximate in a similar manner, simply changing $\mathcal{D} = \mathcal{N}(0, \mathbf{I}_d)$ to $\mathcal{D} = \mathrm{Unif}(\sqrt{d}\mathcal{S}^{d-1})$, a distribution corresponding to Haar measure on the sphere of radius $\sqrt{d}$ in $\mathbb{R}^d$, obtaining estimator $\widehat{\mathrm{SMREG}}_m^+$. As we show in Section 3, such random features can also be used to accurately approximate regular softmax-kernel.

## 2.4 ORTHOGONAL RANDOM FEATURES (ORFs)

The above constitutes the R+ part of the FAVOR+ method. It remains to explain the O-part. To further reduce the variance of the estimator (so that we can use an even smaller number of random features $r$), we entangle different random samples $\omega_1, ..., \omega_m$ to be **exactly** orthogonal. This can be done while maintaining unbiasedness whenever isotropic distributions $\mathcal{D}$ are used (i.e. in particular in all kernels we considered so far) by the standard Gram-Schmidt orthogonalization procedure (see (Choromanski et al., 2017) for details). ORFs is a well-known method, yet it turns out that it works particularly well with our introduced PRFs for softmax. This leads to the **first theoretical results** showing that ORFs can be applied to reduce the variance of softmax/Gaussian kernel estimators **for any** dimensionality $d$ rather than just asymptotically for large enough $d$ (as is the case for previous methods, see: next section) and leads to the **first exponentially small bounds** on large deviations probabilities that are strictly smaller than for non-orthogonal methods. Positivity of random features plays a key role in these bounds. The ORF mechanism requires $m \leq d$, but this will be the case in all our experiments. The pseudocode of the entire FAVOR+ algorithm is given in Appendix B.

Our theoretical results are tightly aligned with experiments. We show in Section 4 that PRFs+ORFs drastically improve accuracy of the approximation of the attention matrix and enable us to reduce $r$ which results in an accurate as well as space and time efficient mechanism which we call FAVOR+.

## 3 THEORETICAL RESULTS

We present here the theory of positive orthogonal random features for softmax-kernel estimation. All these results can be applied also to the Gaussian kernel, since as explained in the previous section, one can be obtained from the other by renormalization (see: Section 2.3). All proofs and additional more general theoretical results with a discussion are given in the Appendix.

**Lemma 2** (positive (hyperbolic) versus trigonometric random features). *The following is true:*

$$\mathrm{MSE}(\widehat{\mathrm{SM}}_m^{\mathrm{trig}}(\mathbf{x}, \mathbf{y})) = \frac{1}{2m} \exp(\|\mathbf{x} + \mathbf{y}\|^2) \mathrm{SM}^{-2}(\mathbf{x}, \mathbf{y})(1 - \exp(-\|\mathbf{x} - \mathbf{y}\|^2))^2,$$

$$\mathrm{MSE}(\widehat{\mathrm{SM}}_m^+(\mathbf{x}, \mathbf{y})) = \frac{1}{m} \exp(\|\mathbf{x} + \mathbf{y}\|^2) \mathrm{SM}^2(\mathbf{x}, \mathbf{y})(1 - \exp(-\|\mathbf{x} + \mathbf{y}\|^2)), \qquad (8)$$

$$\mathrm{MSE}(\widehat{\mathrm{SM}}_m^{\mathrm{hyp}+}(\mathbf{x}, \mathbf{y})) = \frac{1}{2}(1 - \exp(-\|\mathbf{x} + \mathbf{y}\|^2)) \mathrm{MSE}(\widehat{\mathrm{SM}}_m^+(\mathbf{x}, \mathbf{y})),$$

*for independent random samples $\omega_i$, and where* $\mathrm{MSE}$ *stands for the mean squared error.*

Thus, for $\mathrm{SM}(\mathbf{x}, \mathbf{y}) \to 0$ we have: $\mathrm{MSE}(\widehat{\mathrm{SM}}_m^{\mathrm{trig}}(\mathbf{x}, \mathbf{y})) \to \infty$ and $\mathrm{MSE}(\widehat{\mathrm{SM}}_m^+(\mathbf{x}, \mathbf{y})) \to 0$. Furthermore, the hyperbolic estimator provides additional accuracy improvements that are strictly better than those from $\widehat{\mathrm{SM}}_{2m}^+(\mathbf{x}, \mathbf{y})$ with twice as many random features. The next result shows that the regularized softmax-kernel is in practice an accurate proxy of the softmax-kernel in attention.

**Theorem 1** (regularized versus softmax-kernel). *Assume that the $L_\infty$-norm of the attention matrix for the softmax-kernel satisfies: $\|\mathbf{A}\|_\infty \leq C$ for some constant $C \geq 1$. Denote by $\mathbf{A}^{\mathrm{reg}}$ the corresponding attention matrix for the regularized softmax-kernel. The following holds:*

$$\inf_{i,j} \frac{\mathbf{A}^{\mathrm{reg}}(i,j)}{\mathbf{A}(i,j)} \geq 1 - \frac{2}{d^{\frac{1}{3}}} + o\left(\frac{1}{d^{\frac{1}{3}}}\right), \text{ and } \sup_{i,j} \frac{\mathbf{A}^{\mathrm{reg}}(i,j)}{\mathbf{A}(i,j)} \leq 1. \qquad (9)$$

*Furthermore, the latter holds for $d \geq 2$ even if the $L_\infty$-norm condition is not satisfied, i.e. the regularized softmax-kernel is a universal lower bound for the softmax-kernel.*

Consequently, positive random features for SMREG can be used to approximate the softmax-kernel. Our next result shows that orthogonality provably reduces mean squared error of the estimation with positive random features **for any dimensionality** $d > 0$ and we explicitly provide the gap.

**Theorem 2.** *If $\widehat{\mathrm{SM}}_m^{\mathrm{ort}+}(\mathbf{x}, \mathbf{y})$ stands for the modification of $\widehat{\mathrm{SM}}_m^+(\mathbf{x}, \mathbf{y})$ with orthogonal random features (and thus for $m \leq d$), then the following holds for any $d > 0$:*

$$\mathrm{MSE}(\widehat{\mathrm{SM}}_m^{\mathrm{ort}+}(\mathbf{x}, \mathbf{y})) \leq \mathrm{MSE}(\widehat{\mathrm{SM}}_m^+(\mathbf{x}, \mathbf{y})) - \left(1 - \frac{1}{m}\right) \frac{2}{d+2} \mathrm{SM}^2(\mathbf{x}, \mathbf{y}). \qquad (10)$$

*Furthermore, completely analogous result holds for the regularized softmax-kernel* SMREG.

For the regularized softmax-kernel, orthogonal features provide additional concentration results - the first exponentially small bounds for probabilities of estimators' tails that are strictly better than for non-orthogonal variants for every $d > 0$. Our next result enables us to explicitly estimate the gap.

**Theorem 3.** *Let* $\mathbf{x}, \mathbf{y} \in \mathbb{R}^d$. *The following holds for any* $a > \mathrm{SMREG}(\mathbf{x}, \mathbf{y})$ *and* $m \leq d$:

$$\mathbb{P}[\widehat{\mathrm{SMREG}}_m^+(\mathbf{x}, \mathbf{y}) > a] \leq \exp(-m\mathcal{L}_X(a)), \quad \mathbb{P}[\widehat{\mathrm{SMREG}}_m^{\mathrm{ort}+}(\mathbf{x}, \mathbf{y}) > a] \leq \frac{d}{d+2}\exp(-m\mathcal{L}_X(a))$$

*where* $\widehat{\mathrm{SMREG}}_m^{\mathrm{ort}+}(\mathbf{x}, \mathbf{y})$ *stands for the modification of* $\widehat{\mathrm{SMREG}}_m^+(\mathbf{x}, \mathbf{y})$ *with ORFs,* $X = \Lambda \exp(\sqrt{d}\frac{\omega^\top}{\|\omega\|_2}(\mathbf{x} + \mathbf{y}))$, $\omega \sim \mathcal{N}(0, \mathbf{I}_d)$, $\Lambda$ *is as in Lemma 1 and* $\mathcal{L}_Z$ *is a Legendre Transform of* $Z$ *defined as:* $\mathcal{L}_Z(a) = \sup_{\theta > 0} \log(\frac{e^{\theta a}}{M_Z(\theta)})$ *for the moment generating function* $M_Z$ *of* $Z$.

We see that ORFs provide exponentially small and sharper bounds for critical regions where the softmax-kernel is small. Below we show that even for the $\mathrm{SM}^{\mathrm{trig}}$ mechanism with ORFs, it suffices to take $m = \Theta(d \log(d))$ random projections to accurately approximate the attention matrix (thus if not attention renormalization, PRFs would not be needed). In general, $m$ depends on the dimensionality $d$ of the embeddings, radius $R$ of the ball where all queries/keys live and precision parameter $\epsilon$ (see: Appendix F.6 for additional discussion), but does not depend on input sequence length $L$.

**Theorem 4** (uniform convergence for attention approximation). *Assume that* $L_2$-*norms of queries/keys are upper-bounded by* $R > 0$. *Define* $l = Rd^{-\frac{1}{4}}$ *and take* $h^* = \exp(\frac{l^2}{2})$. *Then for any* $\epsilon > 0$, $\delta = \frac{\epsilon}{(h^*)^2}$ *and the number of random projections* $m = \Theta(\frac{d}{\delta^2}\log(\frac{4d^{\frac{3}{4}}R}{\delta}))$ *the following holds for the attention approximation mechanism leveraging estimators* $\widehat{\mathrm{SM}}^{\mathrm{trig}}$ *with ORFs:* $\|\widehat{\mathbf{A}} - \mathbf{A}\|_\infty \leq \epsilon$ *with any constant probability, where* $\widehat{\mathbf{A}}$ *approximates the attention matrix* $\mathbf{A}$.

## 4 EXPERIMENTS

We implemented our setup on top of pre-existing Transformer training code in Jax (Frostig et al., 2018) optimized with just-in-time (`jax.jit`) compilation, and complement our theory with empirical evidence to demonstrate the practicality of FAVOR+ in multiple settings. Unless explicitly stated, a Performer replaces only the attention component with our method, while all other components are exactly the same as for the regular Transformer. For shorthand notation, we denote unidirectional/causal modelling as **(U)** and bidirectional/masked language modelling as **(B)**.

In terms of baselines, we use other Transformer models for comparison, although some of them are restricted to only one case - e.g. Reformer (Kitaev et al., 2020) is only (U), and Linformer (Wang et al., 2020) is only (B). Furthermore, we use PG-19 (Rae et al., 2020) as an alternative (B) pretraining benchmark, as it is made for long-length sequence training compared to the (now publicly unavailable) BookCorpus (Zhu et al., 2015) + Wikipedia dataset used in BERT (Devlin et al., 2018) and Linformer. All model and tokenization hyperparameters are shown in Appendix A.

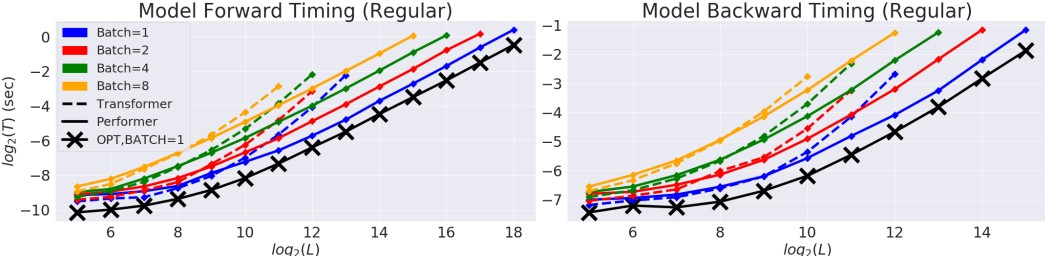

Figure 3: Comparison of Transformer and Performer in terms of forward and backward pass speed and maximum $L$ allowed. "X" (OPT) denotes the maximum possible speedup achievable, when attention simply returns the $\mathbf{V}$-matrix. Plots shown up to when a model produces an out of memory error on a V100 GPU with 16GB. Vocabulary size used was 256. Best in color.

### 4.1 COMPUTATIONAL COSTS

We compared speed-wise the backward pass of the Transformer and the Performer in (B) setting, as it is one of the main computational bottlenecks during training, when using the regular default size $(n_{heads}, n_{layers}, d_{ff}, d) = (8, 6, 2048, 512)$, where $d_{ff}$ denotes the width of the MLP layers. We observed (Fig. 3) that in terms of $L$, the Performer reaches nearly linear time and sub-quadratic

memory consumption (since the explicit $O(L^2)$ attention matrix is not stored). In fact, the Performer achieves nearly optimal speedup and memory efficiency possible, depicted by the "X"-line when attention is replaced with the "identity function" simply returning the $\mathbf{V}$-matrix. The combination of both memory and backward pass efficiencies for large $L$ allows respectively, large batch training and lower wall clock time per gradient step. Extensive additional results are demonstrated in Appendix E by varying layers, raw attention, and architecture sizes.

## 4.2 SOFTMAX ATTENTION APPROXIMATION ERROR

We further examined the approximation error via FAVOR+ in Fig. 4. We demonstrate that **1.** Orthogonal features produce lower error than unstructured (IID) features, **2.** Positive features produce lower error than trigonometric $\sin/\cos$ features. These two empirically validate the PORF mechanism.

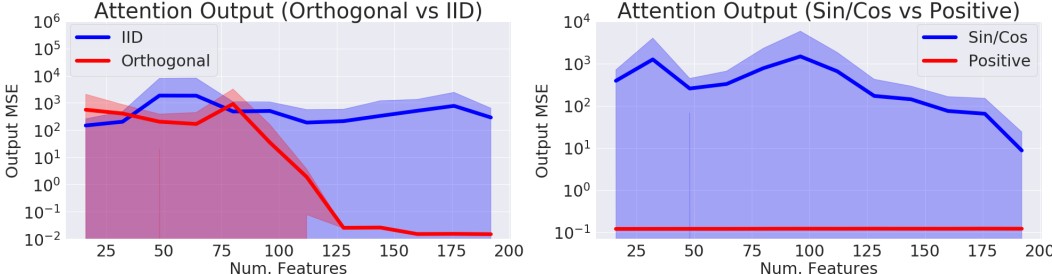

Figure 4: MSE of the approximation output when comparing Orthogonal vs IID features and trigonometric $\sin/\cos$ vs positive features. We took $L = 4096, d = 16$, and varied the number of random samples $m$. Standard deviations shown across 15 samples of appropriately normalized random matrix input data.

To further improve overall approximation of attention blocks across multiple iterations which further improves training, random samples should be periodically redrawn (Fig. 5, right). This is a cheap procedure, but can be further optimized (Appendix B.2).

## 4.3 SOFTMAX APPROXIMATION ON TRANSFORMERS

Even if the approximation of the attention mechanism is tight, small errors can easily propagate throughout multiple Transformer layers (e.g. MLPs, multiple heads), as we show in Fig. 14 (Appendix). In other words, the model's *Lipschitz constant* can easily scale up small attention approximation error, which means that very tight approximations may sometimes be needed. Thus, when applying FAVOR(+)'s softmax approximations on a Transformer model (i.e. "Performer-X-SOFTMAX"), we demonstrate that:

**1.** Backwards compatibility with pretrained models is available as a benefit from softmax approximation, via small finetuning (required due to error propagation) even for trigonometric features (Fig. 5, left) on the LM1B dataset (Chelba et al., 2014). However, when on larger dataset PG-19, **2.** Positive (POS) softmax features (with redrawing) become crucial for achieving performance matching regular Transformers (Fig. 5, right).

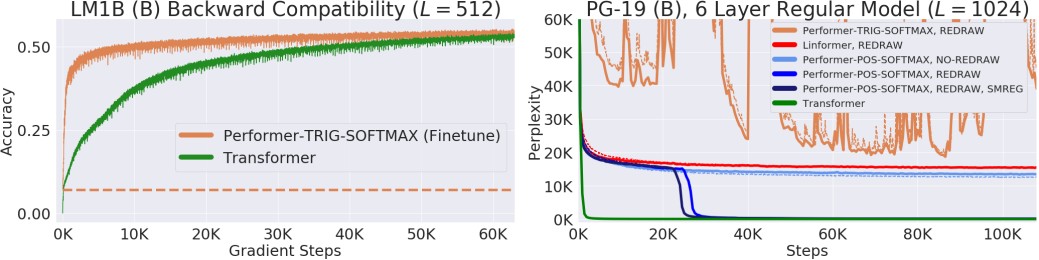

Figure 5: We transferred the original pretrained Transformer's weights into the Performer, which produces an initial non-zero 0.07 accuracy (dotted orange line), but quickly recovers accuracy in a small fraction of the original number of gradient steps. However on PG-19, Trigonometric (TRIG) softmax approximation becomes highly unstable (full curve in Appendix D.2), while positive features (POS) (without redrawing) and Linformer (which also approximates softmax) *even with redrawn projections*, plateau at the same perplexity. Positive softmax with feature redrawing is necessary to match the Transformer, with SMREG (regularization from Sec. 3) allowing faster convergence. Additional ablation studies over many attention kernels, showing also that trigonometric random features lead even to NaN values in training are given in Appendix D.3.

### 4.4 MULTIPLE LAYER TRAINING FOR PROTEINS

We further benchmark the Performer on both (U) and (B) cases by training a 36-layer model using protein sequences from the Jan. 2019 release of TrEMBL (Consortium, 2019), similar to (Madani et al., 2020). In Fig. 6, the Reformer and Linformer *significantly drop in accuracy* on the protein dataset. Furthermore, the usefulness of generalized attention is evidenced by Performer-RELU (taking $f = \mathrm{ReLU}$ in Equation 5) achieving the highest accuracy in both (U) and (B) cases. Our proposed softmax approximation is also shown to be tight, achieving the same accuracy as the exact-softmax Transformer and confirming our theoretical claims from Section 3.

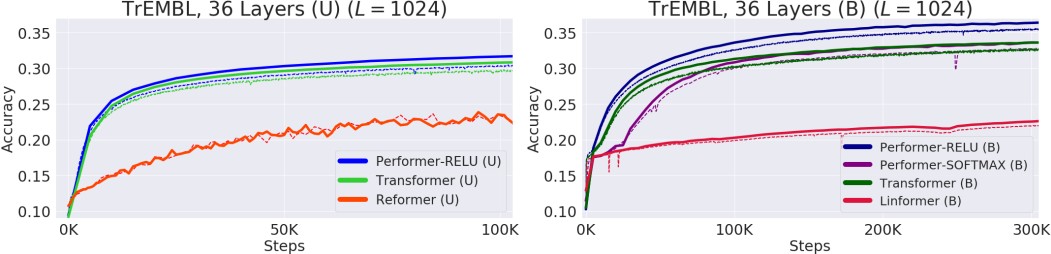

Figure 6: Train = Dashed, Validation = Solid. For TrEMBL, we used the exact same model parameters $(n_{heads}, n_{layers}, d_{ff}, d) = (8, 36, 1024, 512)$ from (Madani et al., 2020) for all runs. For fairness, all TrEMBL experiments used 16x16 TPU-v2's. Batch sizes were maximized for each separate run given the compute constraints. Hyperparameters can be found in Appendix A. Extended results including dataset statistics, out of distribution evaluations, and visualizations, can be found in Appendix C.

### 4.5 LARGE LENGTH TRAINING - COMMON DATASETS

On the standard (U) ImageNet64 benchmark from (Parmar et al., 2018) with $L = 12288$ which is unfeasible for regular Transformers, we set all models to use the same $(n_{heads}, d_{ff}, d)$ but varying $n_{layers}$. Performer/6-layers matches the Reformer/12-layers, while the Performer/12-layers matches the Reformer/24-layers (Fig. 7: left). Depending on hardware (TPU or GPU), we also found that the Performer can be 2x faster than the Reformer via Jax optimizations for the (U) setting.

For a proof of principle study, we also create an initial protein benchmark for predicting interactions among groups of proteins by concatenating protein sequences to length $L = 8192$ from TrEMBL, long enough to model protein interaction networks without the large sequence alignments required by existing methods (Cong et al., 2019). In this setting, a regular Transformer overloads memory even at a batch size of 1 per chip, by a wide margin. Thus as a baseline, we were forced to use a significantly smaller variant, reducing to $(n_{heads}, n_{layers}, d_{ff}, d) = (8, \{1, 2, 3\}, 256, 256)$. Meanwhile, the Performer trains efficiently at a batch size of 8 per chip using the standard $(8, 6, 2048, 512)$ architecture. We see in Fig. 7 (right subfigure) that the smaller Transformer ($n_{layer} = 3$) is quickly bounded at $\approx 19\%$, while the Performer is able to train continuously to $\approx 24\%$.

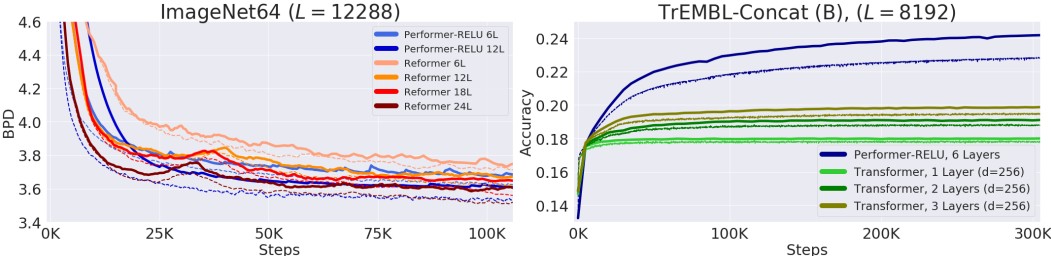

Figure 7: Train = Dashed, Validation = Solid. For ImageNet64, all models used the standard $(n_{heads}, d_{ff}, d) = (8, 2048, 512)$. We further show that our positive softmax approximation achieves the same performance as ReLU in Appendix D.2. For concatenated TrEMBL, we varied $n_{layers} \in \{1, 2, 3\}$ for the smaller Transformer. Hyperparameters can be found in Appendix A.

## 5 CONCLUSION

We presented Performer, a new type of Transformer, relying on our Fast Attention Via positive Orthogonal Random features (FAVOR+) mechanism to significantly improve space and time complexity of regular Transformers. Our mechanism provides to our knowledge the first effective unbiased estimation of the original softmax-based Transformer with linear space and time complexity and opens new avenues in the research on Transformers and the role of non-sparsifying attention mechanisms.

## 6 BROADER IMPACT

We believe that the presented algorithm can be impactful in various ways:

**Biology and Medicine:** Our method has the potential to directly impact research on biological sequence analysis by enabling the Transformer to be applied to much longer sequences without constraints on the structure of the attention matrix. The initial application that we consider is the prediction of interactions between proteins on the proteome scale. Recently published approaches require large evolutionary sequence alignments, a bottleneck for applications to mammalian genomes (Cong et al., 2019). The potentially broad translational impact of applying these approaches to biological sequences was one of the main motivations of this work. We believe that modern bioinformatics can immensely benefit from new machine learning techniques with Transformers being among the most promising. Scaling up these methods to train faster more accurate language models opens the door to the ability to design sets of molecules with pre-specified interaction properties. These approaches could be used to augment existing physics-based design strategies that are of critical importance for example in the development of new nanoparticle vaccines (Marcandalli et al., 2019).

**Environment:** As we have shown, Performers with FAVOR+ are characterized by much lower compute costs and substantially lower space complexity which can be directly translated to $CO_2$ emission reduction (Strubell et al., 2019) and lower energy consumption (You et al., 2020), as regular Transformers require very large computational resources.

**Research on Transformers:** We believe that our results can shape research on efficient Transformers architectures, guiding the field towards methods with strong mathematical foundations. Our research may also hopefully extend Transformers also beyond their standard scope (e.g. by considering the Generalized Attention mechanism and connections with kernels). Exploring scalable Transformer architectures that can handle $L$ of the order of magnitude few thousands and more, preserving accuracy of the baseline at the same time, is a gateway to new breakthroughs in bio-informatics, e.g. language modeling for proteins, as we explained in the paper. Our presented method can be potentially a first step.

**Backward Compatibility:** Our Performer can be used on the top of a regular pre-trained Transformer as opposed to other Transformer variants. Even if up-training is not required, FAVOR+ can still be used for fast inference with no loss of accuracy. We think about this backward compatibility as a very important additional feature of the presented techniques that might be particularly attractive for practitioners.

**Attention Beyond Transformers:** Finally, FAVOR+ can be applied to approximate exact attention also outside the scope of Transformers. This opens a large volume of new potential applications including: hierarchical attention networks (HANS) (Yang et al., 2016), graph attention networks (Velickovic et al., 2018), image processing (Fu et al., 2019), and reinforcement learning/robotics (Tang et al., 2020).

## 7 ACKNOWLEDGEMENTS

We thank Nikita Kitaev and Wojciech Gajewski for multiple discussions on the Reformer, and also thank Aurko Roy and Ashish Vaswani for multiple discussions on the Routing Transformer. We further thank Joshua Meier, John Platt, and Tom Weingarten for many fruitful discussions on biological data and useful comments on this draft. We lastly thank Yi Tay and Mostafa Dehghani for discussions on comparing baselines.

Valerii Likhosherstov acknowledges support from the Cambridge Trust and DeepMind. Lucy Colwell acknowledges support from the Simons Foundation. Adrian Weller acknowledges support from a Turing AI Fellowship under grant EP/V025379/1, The Alan Turing Institute under EPSRC grant EP/N510129/1 and U/B/000074, and the Leverhulme Trust via CFI.

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

APPENDIX: RETHINKING ATTENTION WITH PERFORMERS

## A HYPERPARAMETERS FOR EXPERIMENTS

This optimal setting (including comparisons to approximate softmax) we use for the Performer is specified in the Generalized Attention (Subsec. A.4), and **unless specifically mentioned (e.g. using name "Performer-SOFTMAX"), "Performer" refers to using this generalized attention setting.**

### A.1 METRICS

We report the following evaluation metrics:

1. **Accuracy**: For unidirectional models, we measure the accuracy on next-token prediction, averaged across all sequence positions in the dataset. For bidirectional models, we mask each token with $15\%$ probability (same as (Devlin et al., 2018)) and measure accuracy across the masked positions.

2. **Perplexity**: For unidirectional models, we measure perplexity across all sequence positions in the dataset. For bidirectional models, similar to the accuracy case, we measure perplexity across the masked positions.

3. **Bits Per Dimension/Character (BPD/BPC)**: This calculated by loss divided by $\ln(2)$.

We used the full evaluation dataset for TrEMBL in the plots in the main section, while for other datasets such as ImageNet64 and PG-19 which have very large evaluation dataset sizes, we used random batches (>2048 samples) for plotting curves.

#### A.1.1 PG-19 PREPROCESSING

The PG-19 dataset (Rae et al., 2020) is presented as a challenging long range text modeling task. It consists of out-of-copyright Project Gutenberg books published before 1919. It does not have a fixed vocabulary size, instead opting for any tokenization which can model an arbitrary string of text. We use a unigram SentencePiece vocabulary (Kudo & Richardson, 2018) with 32768 tokens, which maintains whitespace and is completely invertible to the original book text. Perplexities are calculated as the average log-likelihood per token, multiplied by the ratio of the sentencepiece tokenization to number of tokens in the original dataset. The original dataset token count per split is: train=1973136207, validation=3007061, test=6966499. Our sentencepiece tokenization yields the following token counts per split: train=3084760726, valid=4656945, and test=10699704. This gives log likelihood multipliers of train=1.5634, valid=1.5487, test=1.5359 per split before computing perplexity, which is equal to $\exp(\text{log likelihood multiplier} * \text{loss})$.

Preprocessing for TrEMBL is extensively explained in Appendix C.

### A.2 TRAINING HYPERPARAMETERS

Unless specifically stated, all Performer + Transformer runs by default used $0.5$ grad clip, $0.1$ weight decay, $0.1$ dropout, $10^{-3}$ fixed learning rate with Adam hyperparameters ($\beta_1 = 0.9, \beta_2 = 0.98, \epsilon = 10^{-9}$), with batch size maximized (until TPU memory overload) for a specific model.

All 36-layer protein experiments used the same amount of compute (i.e. 16x16 TPU-v2, 8GB per chip). For concatenated experiments, 16x16 TPU-v2's were also used for the Performer, while 8x8's were used for the 1-3 layer ($d = 256$) Transformer models (using 16x16 did not make a difference in accuracy).

**Note that Performers are using the same training hyperparameters as Transformers, yet achieving competitive results** - this shows that FAVOR can act as a simple drop-in without needing much tuning.

### A.3 APPROXIMATE SOFTMAX ATTENTION DEFAULT VALUES

The optimal values, set to default parameters[1], are: renormalize_attention = True, numerical stabilizer = $10^{-6}$, number of features = 256, ortho_features = True, ortho_scaling = 0.0.

---

[1]`https://github.com/google-research/google-research/blob/master/performer/fast_attention`

## A.4 GENERALIZED ATTENTION DEFAULT VALUES

The optimal values, set to default parameters[2] , are: renormalize_attention = True, numerical stabilizer = 0.0, number of features = 256, kernel = ReLU, kernel_epsilon = $10^{-3}$.

## A.5 REFORMER DEFAULT VALUES

For the Reformer, we used the same hyperparameters as mentioned for protein experiments, without gradient clipping, while using the defaults[3] (which instead use learning rate decay) for ImageNet-64. In both cases, the Reformer used the same default LSH attention parameters.

## A.6 LINFORMER DEFAULT VALUES

Using our standard pipeline as mentioned above, we replaced the attention function with the Linformer variant via Jax, with $\delta = 10^{-6}, k = 600$ (same notation used in the paper (Wang et al., 2020)), where $\delta$ is the exponent in a renormalization procedure using $e^{-\delta}$ as a multiplier in order to approximate softmax, while $k$ is the dimension of the projections of the $\mathbf{Q}$ and $\mathbf{K}$ matrices. As a sanity check, we found that our Linformer implementation in Jax correctly approximated exact softmax's output within $0.02$ error for all entries.

Note that for rigorous comparisons, our Linformer hyperparameters are even stronger than the defaults found in (Wang et al., 2020), as:

- We use $k = 600$, which is more than twice than the default $k = 256$ from the paper, and also twice than our default $m = 256$ number of features.
- We also use redrawing, which avoids "unlucky" projections on $\mathbf{Q}$ and $\mathbf{K}$.

---

[2]https://github.com/google-research/google-research/blob/master/performer/fast_attention
[3]https://github.com/google/trax/blob/master/trax/supervised/configs/reformer_imagenet64.gin

## B  MAIN ALGORITHM: FAVOR+

We outline the main algorithm for FAVOR+ formally:

---

**Algorithm 1:** FAVOR+ (bidirectional or unidirectional).

---

**Input :** $\mathbf{Q}, \mathbf{K}, \mathbf{V} \in \mathbb{R}^{L \times d}$, isBidirectional - binary flag.
**Result:** $\widehat{\mathrm{Att}}_{\leftrightarrow}(\mathbf{Q}, \mathbf{K}, \mathbf{V}) \in \mathbb{R}^{L \times L}$ if isBidirectional, $\widehat{\mathrm{Att}}_{\rightarrow}(\mathbf{Q}, \mathbf{K}, \mathbf{V}) \in \mathbb{R}^{L \times L}$ otherwise.
Compute $\mathbf{Q}'$ and $\mathbf{K}'$ as described in Section 2.2 and Section 2.3 and take $\mathbf{C} := [\mathbf{V} \quad \mathbf{1}_L]$;
**if** isBidirectional **then**
  $\text{Buf}_1 := (\mathbf{K}')^\top \mathbf{C} \in \mathbb{R}^{M \times (d+1)}, \quad \text{Buf}_2 := \mathbf{Q}' \text{Buf}_1 \in \mathbb{R}^{L \times (d+1)}$;
**else**
  Compute $\mathbf{G}$ and its prefix-sum tensor $\mathbf{G}^{\mathrm{PS}}$ according to (11);
  $\text{Buf}_2 := \left[ \mathbf{G}^{\mathrm{PS}}_{1,:,:} \mathbf{Q}'_1 \quad \cdots \quad \mathbf{G}^{\mathrm{PS}}_{L,:,:} \mathbf{Q}'_L \right]^\top \in \mathbb{R}^{L \times (d+1)}$;
**end**
$[\text{Buf}_3 \quad \text{buf}_4] := \text{Buf}_2, \quad \text{Buf}_3 \in \mathbb{R}^{L \times d}, \quad \text{buf}_4 \in \mathbb{R}^L$;
**return** $\mathrm{diag}(\text{buf}_4)^{-1} \text{Buf}_3$;

---

### B.1  UNIDIRECTIONAL CASE AND PREFIX SUMS

We explain how our analysis from Section 2.2 can be extended to the unidirectional mechanism in this section. Notice that this time attention matrix $\mathbf{A}$ is masked, i.e. all its entries not in the lower-triangular part (which contains the diagonal) are zeroed (see also Fig. 8).

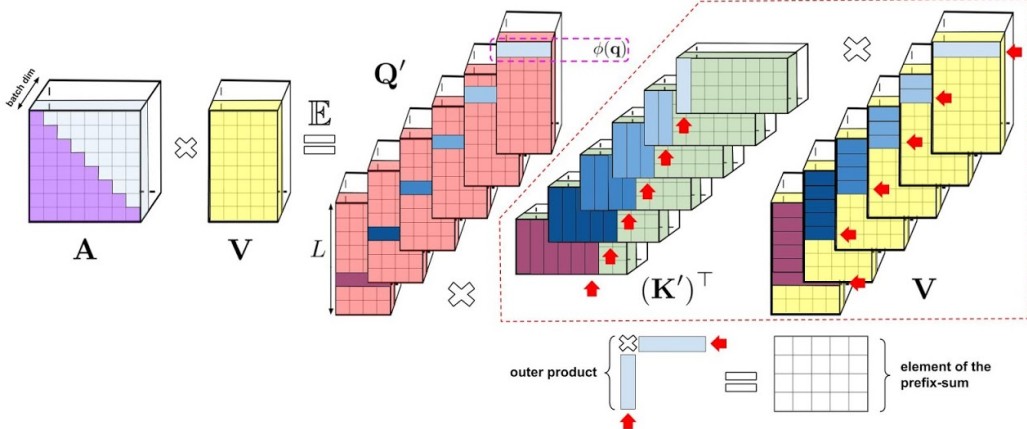

Figure 8: Visual representation of the prefix-sum algorithm for unidirectional attention. For clarity, we omit attention normalization in this visualization. The algorithm keeps the prefix-sum which is a matrix obtained by summing the outer products of random features corresponding to keys with value-vectors. At each given iteration of the prefix-sum algorithm, a random feature vector corresponding to a query is multiplied by the most recent prefix-sum (obtained by summing all outer-products corresponding to preceding tokens) to obtain a new row of the matrix $\mathbf{AV}$ which is output by the attention mechanism.

For the unidirectional case, our analysis is similar as for the bidirectional case, but this time our goal is to compute $\mathrm{tril}(\mathbf{Q}'(\mathbf{K}')^\top)\mathbf{C}$ without constructing and storing the $L \times L$-sized matrix $\mathrm{tril}(\mathbf{Q}'(\mathbf{K}')^\top)$ explicitly, where $\mathbf{C} = [V \quad \mathbf{1}_L] \in \mathbb{R}^{L \times (d+1)}$. In order to do so, observe that $\forall 1 \le i \le L$:

$$[\mathrm{tril}(\mathbf{Q}'(\mathbf{K}')^\top)\mathbf{C}]_i = \mathbf{G}^{\mathrm{PS}}_{i,:,:} \times \mathbf{Q}'_i, \quad \mathbf{G}^{\mathrm{PS}}_{i,:,:} = \sum_{j=1}^{i} \mathbf{G}_{j,:,:}, \quad \mathbf{G}_{j,:,:} = \mathbf{K}'_j \mathbf{C}_j^\top \in \mathbb{R}^{M \times (d+1)} \quad (11)$$

where $\mathbf{G}, \mathbf{G}^{\mathrm{PS}} \in \mathbb{R}^{L \times M \times (d+1)}$ are 3d-tensors. Each slice $\mathbf{G}^{\mathrm{PS}}_{:,l,p}$ is therefore a result of a prefix-sum (or cumulative-sum) operation applied to $\mathbf{G}_{:,l,p}$: $\mathbf{G}^{\mathrm{PS}}_{i,l,p} = \sum_{j=1}^{i} \mathbf{G}_{i,l,p}$. An efficient algorithm to compute the prefix-sum of $L$ elements takes $O(L)$ total steps and $O(\log L)$ time when computed in parallel (Ladner & Fischer, 1980; Cormen et al., 2009). See Algorithm 1 for the whole approach.

### B.2  ORTHOGONAL RANDOM FEATURES - EXTENSIONS

As mentioned in the main text, for isotropic $\Omega$ (true for most practical applications, including regular attention), instead of sampling $\omega_i$ independently, we can use *orthogonal random features* (ORF) (Yu

et al., 2016; Choromanski et al., 2017; 2018b): these maintain the marginal distributions of samples $\omega_i$ while enforcing that different samples are orthogonal. If we need $m > d$, ORFs still can be used locally within each $d \times d$ block of $\mathbf{W}$ (Yu et al., 2016).

ORFs were introduced to reduce the variance of Monte Carlo estimators (Yu et al., 2016; Choromanski et al., 2017; 2018b; 2019a; Rowland et al., 2019; Choromanski et al., 2018a; 2019b) and we showed in the theoretical and experimental sections from the main body that they do indeed lead to more accurate approximations and substantially better downstream results. There exist several variants of the ORF-mechanism and in the main body we discussed only the base one (that we refer to here as *regular*). Below we briefly review the most efficient ORF mechanisms (based on their strengths and costs) to present the most complete picture.

**(1) Regular ORFs [R-ORFs]:** Applies Gaussian orthogonal matrices (Yu et al., 2016). Encodes matrix $\mathbf{W}$ of $\omega$-samples (with different rows corresponding to different samples) in $O(md)$ space. Provides algorithm for computing $\mathbf{Wx}$ in $O(md)$ time for any $\mathbf{x} \in \mathbb{R}^d$. Gives unbiased estimation. Requires one-time $O(md^2)$ preprocessing (Gram-Schmidt orthogonalization).

**(2) Hadamard/Givens ORFs [H/G-ORFs]:** Applies random Hadamard (Choromanski et al., 2017) or Givens matrices (Choromanski et al., 2019b). Encodes matrix $\mathbf{W}$ in $O(m)$ or $O(m \log(d))$ space. Provides algorithm for computing $\mathbf{Wx}$ in $O(m \log(d))$ time for any $\mathbf{x} \in \mathbb{R}^d$. Gives small bias (tending to 0 with $d \to \infty$).

### B.3    Time and Space Complexity - Detailed Analysis

We see that a variant of bidirectional FAVOR+ using iid samples or R-ORFs has $O(md + Ld + mL)$ space complexity as opposed to $\Theta(L^2 + Ld)$ space complexity of the baseline. Unidirectional FAVOR+ using fast prefix-sum pre-computation in parallel (Ladner & Fischer, 1980; Cormen et al., 2009) has $O(mLd)$ space complexity to store $\mathbf{G}^{\text{PS}}$ which can be reduced to $O(md + Ld + mL)$ by running a simple (though non-parallel in $L$) aggregation of $\mathbf{G}^{\text{PS}}_{i,:,:}$ without storing the whole tensor $\mathbf{G}^{\text{PS}}$ in memory. From Subsec. B.2, we know that if instead we use G-ORFs, then space complexity is reduced to $O(m \log(d) + Ld + mL)$ and if the H-ORFs mechanism is used, then space is further reduced to $O(m + Ld + mL) = O(Ld + mL)$. Thus for $m, d \ll L$ all our variants provide substantial space complexity improvements since they do not need to store the attention matrix explicitly.

The time complexity of Algorithm 1 is $O(Lmd)$ (note that constructing $\mathbf{Q}'$ and $\mathbf{K}'$ can be done in time $O(Lmd)$). Note that the time complexity of our method is much lower than $O(L^2 d)$ of the baseline for $L \gg m$.

As explained in Subsec. B.2, the R-ORF mechanism incurs an extra one-time $O(md^2)$ cost (negligible compared to the $O(Lmd)$ term for $L \gg d$). H-ORFs or G-ORFs do not have this cost, and when FAVOR+ uses them, computing $\mathbf{Q}'$ and $\mathbf{K}'$ can be conducted in time $O(L \log(m)d)$ as opposed to $O(Lmd)$ (see: Subsec. B.2). Thus even though H/G-ORFs do not change the asymptotic time complexity, they improve the constant factor from the leading term. This might play an important role in training very large models.

The number of random features $m$ allows a trade-off between computational complexity and the level of approximation: bigger $m$ results in higher computation costs, but also in a lower variance of the estimate of $\mathbf{A}$. In the theoretical section from the main body we showed that in practice we can take $M = \Theta(d \log(d))$.

Observe that the FAVOR+ algorithm is highly-parallelizable, and benefits from fast matrix multiplication and broadcasted operations on GPUs or TPUs.

## C    EXPERIMENTAL DETAILS FOR PROTEIN MODELING TASKS

### C.1    TREMBL DATASET

| Dataset | Set Name | Count | Length Statistics | | | | |
|---|---|---|---|---|---|---|---|
| | | | Min | Max | Mean | STD | Median |
| TrEMBL | Train | 104,863,744 | 2 | 74,488 | 353.09 | 311.16 | 289.00 |
| | Valid | 102,400 | 7 | 11,274 | 353.62 | 307.42 | 289.00 |
| | Test | 1,033,216 | 8 | 32,278 | 353.96 | 312.23 | 289.00 |
| | OOD | 29,696 | 24 | 4,208 | 330.96 | 269.86 | 200.00 |
| TrEMBL (concat) | Train | 4,532,224 | 8,192 | 8,192 | 8,192 | 0 | 8,192 |
| | Valid | 4,096 | | | | | |

Table 1: Statistics for the TrEMBL single sequence and the long sequence task.

We used the TrEMBL dataset[4], which contains 139,394,261 sequences of which 106,030,080 are unique. While the training dataset appears smaller than the one used in Madani et al. (Madani et al., 2020), we argue that it includes most of the relevant sequences. Specifically, the TrEMBL dataset consists of the subset of UniProtKB sequences that have been computationally analyzed but not manually curated, and accounts for $\approx 99.5\%$ of the total number of sequences in the UniProtKB dataset[5].

Following the methodology described in Madani et al. (Madani et al., 2020), we used both an OOD-Test set, where a selected subset of Pfam families are held-out for valuation, and an IID split, where the remaining protein sequences are split randomly into train, valid, and test tests. We held-out the following protein families (PF18369, PF04680, PF17988, PF12325, PF03272, PF03938, PF17724, PF10696, PF11968, PF04153, PF06173, PF12378, PF04420, PF10841, PF06917, PF03492, PF06905, PF15340, PF17055, PF05318), which resulted in 29,696 OOD sequences. We note that, due to deduplication and potential TrEMBL version mismatch, our OOD-Test set does not match exactly the one in Madani et al. (Madani et al., 2020). We also note that this OOD-Test selection methodology does not guarantee that the evaluation sequences are within a minimum distance from the sequences used during training. In future work, we will include rigorous distance based splits.

The statistics for the resulting dataset splits are reported in Table 1. In the standard sequence modeling task, given the length statistics that are reported in the table, we clip single sequences to maximum length $L = 1024$, which results in few sequences being truncated significantly.

In the long sequence task, the training and validation sets are obtained by concatenating the sequences, separated by an end-of-sequence token, and grouping the resulting chain into non-overlapping sequences of length $L = 8192$.

### C.2    EMPIRICAL BASELINE

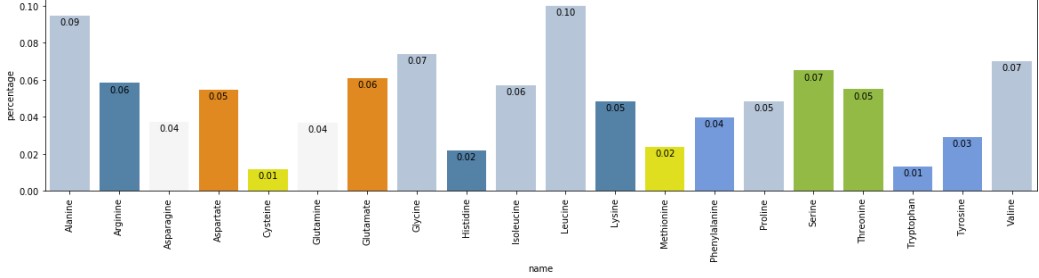

Figure 9: Visualization of the estimated empirical distribution for the 20 standard amino acids, colored by their class. Note the consistency with the statistics on the TrEMBL web page.

A random baseline, with uniform probability across all the vocabulary tokens at every position, has accuracy $5\%$ (when including only the 20 standard amino acids) and $4\%$ (when also including the 5 anomalous amino acids (Consortium, 2019)). However, the empirical frequencies of the various

---

[4]https://www.uniprot.org/statistics/TrEMBL
[5]https://www.uniprot.org/uniprot/

amino acids in our dataset may be far from uniform, so we also consider an *empirical baseline* where the amino acid probabilities are proportional to their empirical frequencies in the training set.

Figure 9 shows the estimated empirical distribution. We use both the standard and anomalous amino acids, and we crop sequences to length 1024 to match the data processing performed for the Transformer models. The figure shows only the 20 standard amino acids, colored by their class, for comparison with the visualization on the TrEMBL web page[6].

## C.3 Tabular Results

Table 2 contains the results on the single protein sequence modeling task ($L = 1024$). We report accuracy and perplexity as defined in Appendix A:

| Model Type | Set Name | Model | Accuracy | Perplexity |
|---|---|---|---|---|
| UNI | Test | Empirical Baseline | 9.92 | 17.80 |
| | | Transformer | 30.80 | 9.37 |
| | | Performer (generalized) | 31.58 | 9.17 |
| | OOD | Empirical Baseline | 9.07 | 17.93 |
| | | Transformer | 19.70 | 13.20 |
| | | Performer (generalized) | 18.44 | 13.63 |
| BID | Test | Transformer | 33.32 | 9.22 |
| | | Performer (generalized) | 36.09 | 8.36 |
| | | Performer (softmax) | 33.00 | 9.24 |
| | OOD | Transformer | 25.07 | 12.09 |
| | | Performer (generalized) | 24.10 | 12.26 |
| | | Performer (softmax) | 23.48 | 12.41 |

Table 2: Results on single protein sequence modeling ($L = 1024$). We note that the empirical baseline results are applicable to both the unidirectional (UNI) and bidirectional (BID) models.

## C.4 Attention Matrix Illustration

In this section we illustrate the attention matrices produced by a Performer model. We focus on the bidirectional case and choose one Performer model trained on the standard single-sequence TrEMBL task for over 500K steps. The same analysis can be applied to unidirectional Performers as well.

We note that while the Transformer model instantiates the attention matrix in order to compute the attention output that incorporates the (queries $Q$, keys $K$, values $V$) triplet (see Eq. 1 in the main paper), the FAVOR mechanism returns the attention output directly (see Algorithm 1). To account for this discrepancy, we extract the attention matrices by applying each attention mechanism twice: once on each original $(Q, K, V)$ triple to obtain the attention output, and once on a modified $(Q, K, V^\circ)$ triple, where $V^\circ$ contains one-hot indicators for each position index, to obtain the attention matrix. The choice of $V^\circ$ ensures that the dimension of the attention output is equal to the sequence length, and that a non-zero output on a dimension $i$ can only arise from a non-zero attention weight to the $i^{th}$ sequence position. Indeed, in the Transformer case, when comparing the output of this procedure with the instantiated attention matrix, the outputs match.

**Attention matrix example.** We start by visualizing the attention matrix for an individual protein sequence. We use the BPT1_BOVIN protein sequence[7], one of the most extensively studied globular proteins, which contains 100 amino acids. In Figure 10, we show the attention matrices for the first 4 layers. Note that many heads show a *diagonal* pattern, where each node attends to its neighbors, and some heads show a *vertical* pattern, where each head attends to the same fixed positions. These patterns are consistent with the patterns found in Transformer models trained on natural language

---

[6]https://www.uniprot.org/statistics/TrEMBL
[7]https://www.uniprot.org/uniprot/P00974

(Kovaleva et al., 2019). In Figure 12 we highlight these attention patterns by focusing on the first 25 tokens, and in Figure 11, we illustrate in more detail two attention heads.

**Amino acid similarity.** Furthermore, we analyze the amino-acid similarity matrix estimated from the attention matrices produced by the Performer model, as described in Vig et al. (Vig et al., 2020). We aggregate the attention matrix across 800 sequences. The resulting similarity matrix is illustrated in Figure 13. Note that the Performer recognizes highly similar amino acid pairs such as (D, E) and (F, Y).

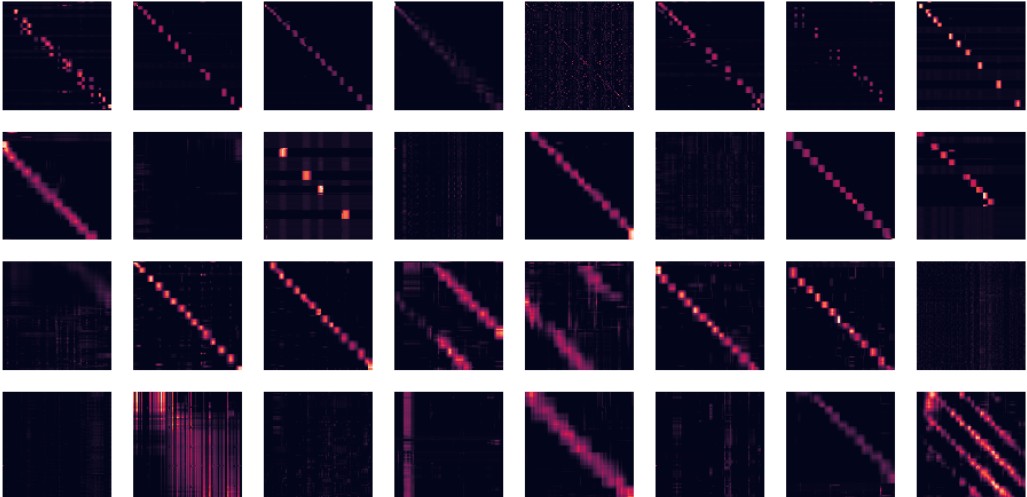

Figure 10: We show the attention matrices for the first 4 layers and all 8 heads (each row is a layer, each column is head index, each cell contains the attention matrix across the entire BPT1_BOVIN protein sequence). Note that many heads show a *diagonal* pattern, where each node attends to its neighbors, and some heads show a *vertical* pattern, where each head attends to the same fixed positions.

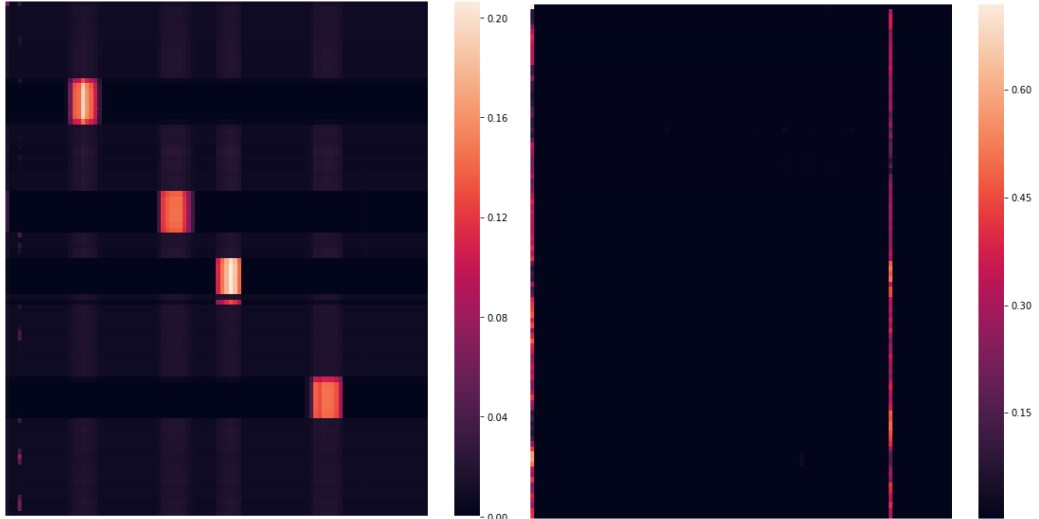

Figure 11: We illustrate in more detail two attention heads. The sub-figures correspond respectively to: **(1)** Head 1-2 (second layer, third head), **(2)** Head 4-1 (fifth layer, second head). Note the block attention in Head 1-2 and the vertical attention (to the start token ('M') and the 85th token ('C')) in Head 4-1.

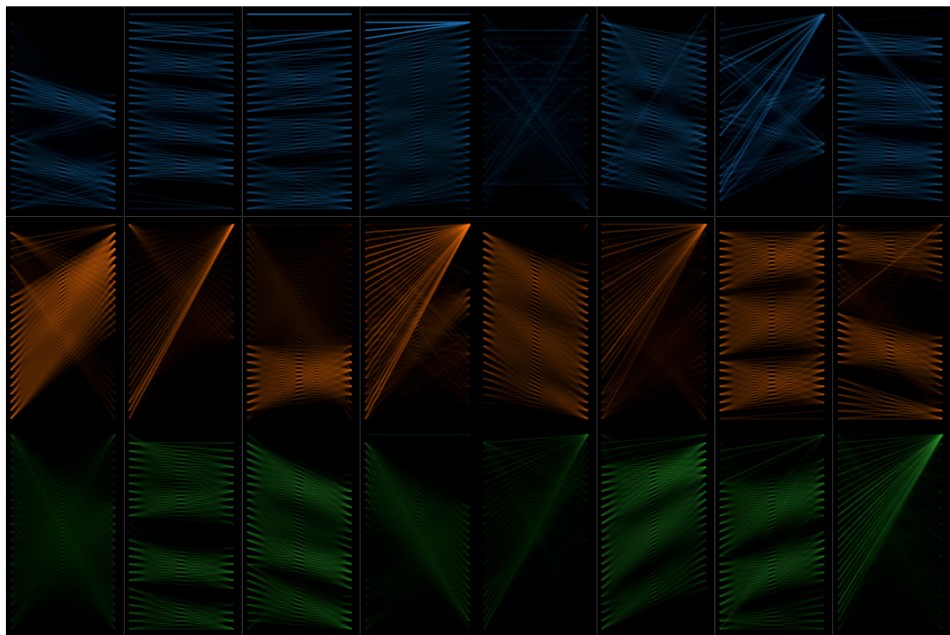

Figure 12: We highlight the attention patterns by restricting our attention to the first 25 tokens (note that we do not renormalize the attention to these tokens). The illustration is based on Vig et al. (Vig, 2019; Vig & Belinkov, 2019). Note that, similar to prior work on protein Transformers (Madani et al., 2020), the attention matrices include both local and global patterns.

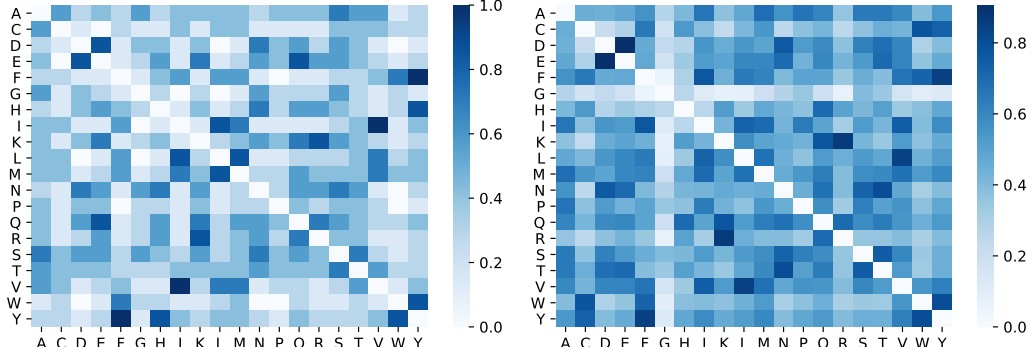

Figure 13: Amino acid similarity matrix estimated from attention matrices aggregated across a small subset of sequences, as described in Vig et al. (Vig et al., 2020). The sub-figures correspond respectively to: **(1)** the normalized BLOSUM matrix, **(2)** the amino acid similarity estimated via a trained Performer model. Note that the Performer recognizes highly similar amino acid pairs such as (D, E) and (F, Y).

# D EXTENDED APPROXIMATION AND COMPARISON RESULTS

## D.1 BACKWARDS COMPATIBILITY - ERROR PROPAGATION

Although mentioned previously (Sec. 4.2) that the Performer with additional finetuning is backwards compatible with the Transformer, we demonstrate below in Fig. 14 that error propagation due to non-attention components of the Transformer is one of the primary reasons that pretrained Transformer weights cannot be immediately used for inference on the corresponding Performer.

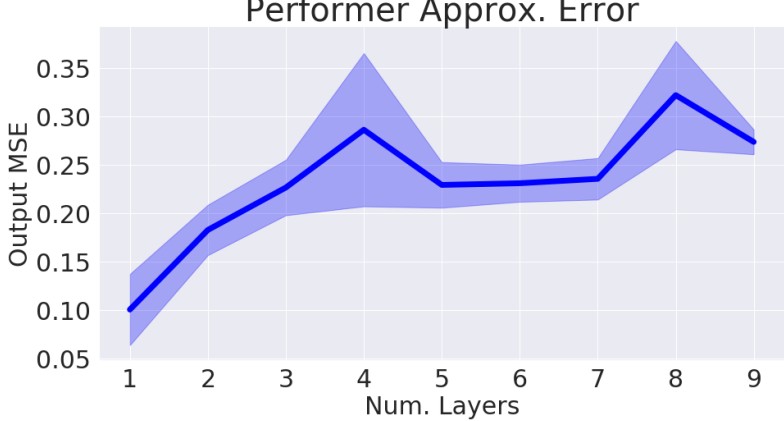

Figure 14: Output approximation errors between a vanilla Transformer and a Performer (with orthogonal features) for varying numbers of layers.

## D.2 APPROXIMATE SOFTMAX - EXTENDED PROPERTIES

We show the following properties of our softmax approximation, in Fig. 15:

**Redrawing:** While the benefits of redrawing features was shown in Subsec. 4.3 of the main body of the paper, we also its benefits when there are multiple layers with large scale (16x16 TPU-v2) training.

**Unidirectional:** While we have shown on TrEMBL that Performer with generalized ReLU attention outperforms softmax, we also show that approximate softmax attention can still be a solid choice, for example on ImageNet64 (U). After 100K steps of training, the Performer-ReLU, Performer-Softmax, and Performer-Softmax (SMREG) variants achieve respectively, 3.67, 3.69, 3.67 BPD.

**Instability of Trigonometric Features:** We see the full view of the unstable training curve when using Trigonometric softmax.

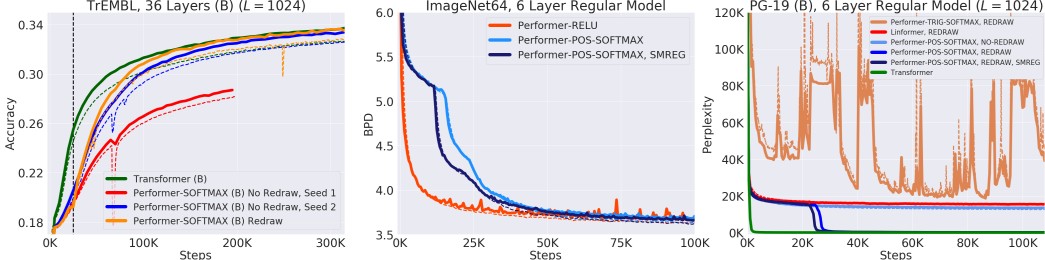

Figure 15: Best viewed zoomed in. **Left:** The importance of redrawing features. If redrawing is not used, an "unlucky" set of random features may cause training degradation, shown by the early-stopped curve with Seed 1, while a 'lucky' set of random features may cause no issue, shown by the curve with Seed 2. Redrawing allows the training to correct itself, as seen at the black vertical line. **Middle:** Using the same 8x8 TPU-v2 compute and same 6-layer standard model, approximate softmax with positive features achieves the same result as generalized ReLU attention. **Right:** Zoomed out view of right subfigure of Fig. 5, showing that Trigonometric softmax causes very unstable training behaviors.

## D.3 GENERALIZED ATTENTION

We investigated Generalized Attention mechanisms (mentioned in Sec. 2.2) on TrEMBL when $L = 512$ for various kernel functions. This is similar to (Tsai et al., 2019) which also experiments with various attention kernels for natural language. Using hyperparameter sweeps across multiple

variables in FAVOR, we compared several kernels and also renormalization on/off (Fig. 16 and Fig. 17), where Renormalize corresponds to applying $\mathbf{D}^{-1}$ operator in attention, as for the standard mechanism, though we noticed that disabling it does not necessarily hurt accuracy) to produce the best training configuration for the Performer. We note that the effective batch size slightly affects the rankings (as shown by the difference between 2x2 and 4x4 TPU runs) - we by default use the generalized ReLU kernel with other default hyperparameters shown in Appendix A, as we observed that they are empirically optimal for large batch size runs (i.e. 8x8 or 16x16 TPU's).

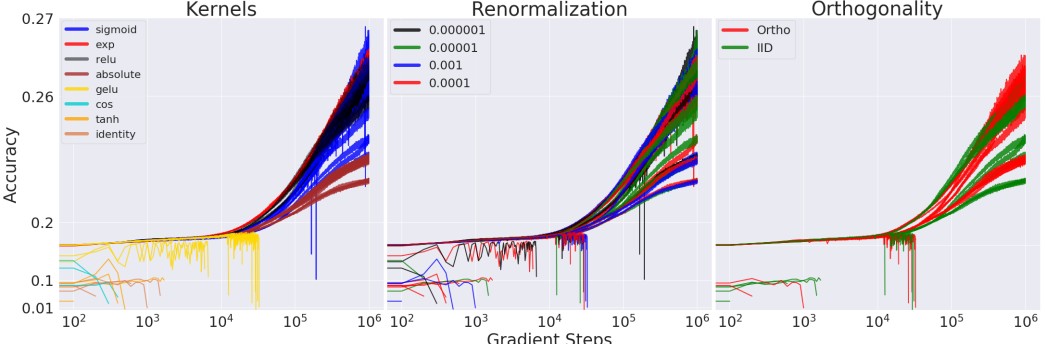

Figure 16: To emphasize the highest accuracy runs but also show the NaN issues with certain kernels which caused runs to stop early, we set both x and y axes to be log-scale. We tested kernels defined by different functions $f$ (see: Sec. 2.2): sigmoid, exponential, ReLU, absolute, gelu, cosine (original softmax approximation), tanh, and identity. All training runs were performed on 2x2 TPU-v2's, 128 batch size per device.

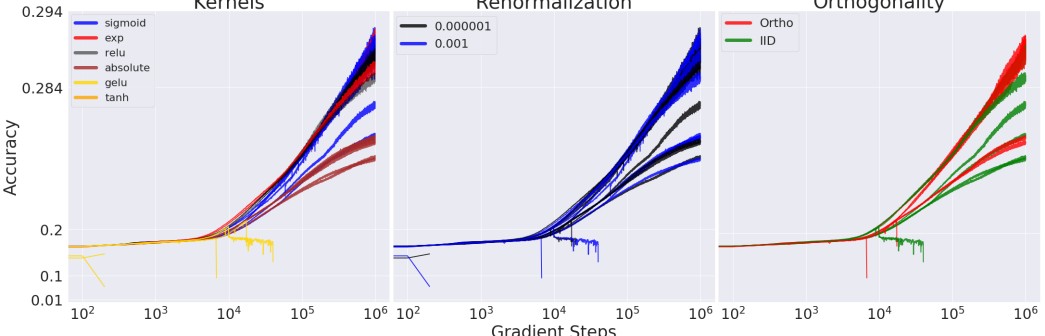

Figure 17: We also performed a similar setup as Fig. 16 for 4x4 TPU-v2's.

### D.4 COMPARISON WITH LINEAR TRANSFORMER

We use the attention implementation of the Linear Transformer from (Katharopoulos et al., 2020), which mainly involves setting our feature map $\phi(x) = \text{elu}(x) + 1$, where $\text{elu}(x)$ is the shifted-eLU function from (Clevert et al., 2016).

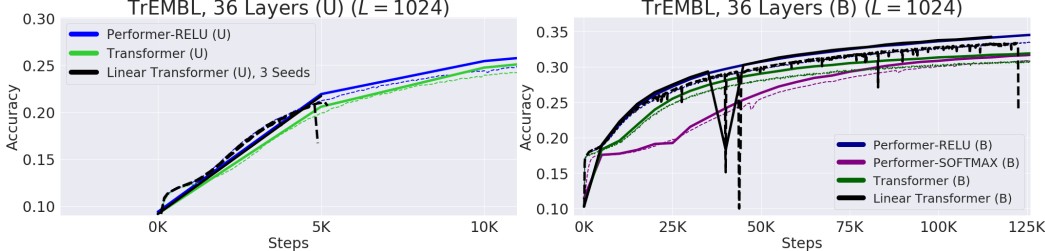

Figure 18: **Left:** In the unidirectional 36-ProGen setting, we ran 3 seeds of the Linear Transformer, and found that all 3 seeds produced exploding gradients very early on, stopping the training run. **Right:** The Linear Transformer in the bidirectional setting also produced an exploding gradient in the middle of training, near 125K steps. Exploding gradients can be evidenced by the sharp drop in train accuracy right before a NaN error.

For the sake of fairness and to prevent confounding results, while (Katharopoulos et al., 2020) also uses the GeLU nonlinearity for the MLPs in the Linear Transformer, we instead use the original ReLU nonlinearity. We also used the exact same training hyperparameters as Performer-ReLU on

our exact ProGen setting from Fig. 6. Ultimately, we empirically found that the Linear Transformer possessed numerical instability during training via unstable training curves, **ultimately stopping training by producing exploding gradients (NaNs)** (Fig. 18).

### D.5 LONG RANGE ARENA

Performers are compared against many additional (scalable and not scalable) methods not included in our paper: *Local Attention*, *Sparse Attention*, *Longformer*, *Sinkhorn Transformer*, *Synthesizer*, *Big Bird* and the aforementioned *Linear Transformer* on challenging long range context tasks in the Long Range Arena (Tay et al., 2021), with Fig. 19 displaying the original paper's results. Performers obtain the largest LRA (Long Range Arena) score among all tested **scalable** Transformers methods (which we define by having speed of > 100 examples/sec).

Tasks used for comparison include: **(1)** a longer variation of the standard ListOps task proposed in (Nangia & Bowman, 2018), **(2)** byte-level text classification using real-world data, **(3)** byte-level document retrieval, **(4)** image classification on sequences of pixels, and **(5)** Pathfinder task (long-range spatial dependency problem). In the Long Range Arena paper, the authors found that all models do not learn anything on Path-X task (denoted by FAIL), contrary to the Pathfinder task, which shows that increasing the sequence length can cause seriously difficulties for model training.

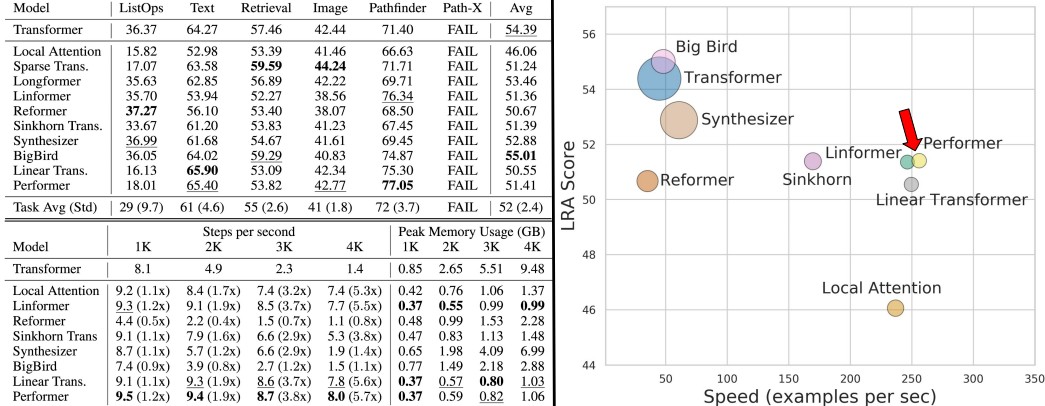

| Model | ListOps | Text | Retrieval | Image | Pathfinder | Path-X | Avg |
|---|---|---|---|---|---|---|---|
| Transformer | 36.37 | 64.27 | 57.46 | 42.44 | 71.40 | FAIL | 54.39 |
| Local Attention | 15.82 | 52.98 | 53.39 | 41.46 | 66.63 | FAIL | 46.06 |
| Sparse Trans. | 17.07 | 63.58 | **59.59** | **44.24** | 71.71 | FAIL | 51.24 |
| Longformer | 35.63 | 62.85 | 56.89 | 42.22 | 69.71 | FAIL | 53.46 |
| Linformer | 35.70 | 53.94 | 52.27 | 38.56 | 76.34 | FAIL | 51.36 |
| Reformer | **37.27** | 56.10 | 53.40 | 38.07 | 68.50 | FAIL | 50.67 |
| Sinkhorn Trans. | 33.67 | 61.20 | 53.83 | 41.23 | 67.45 | FAIL | 51.39 |
| Synthesizer | 36.99 | 61.68 | 54.67 | 41.61 | 69.45 | FAIL | 52.88 |
| BigBird | 36.05 | 64.02 | 59.29 | 40.83 | 74.87 | FAIL | **55.01** |
| Linear Trans. | 16.13 | **65.90** | 53.09 | 42.34 | 75.30 | FAIL | 50.55 |
| Performer | 18.01 | 65.40 | 53.82 | 42.77 | **77.05** | FAIL | 51.41 |
| Task Avg (Std) | 29 (9.7) | 61 (4.6) | 55 (2.6) | 41 (1.8) | 72 (3.7) | FAIL | 52 (2.4) |

| Model | Steps per second | | | | Peak Memory Usage (GB) | | | |
|---|---|---|---|---|---|---|---|---|
| | 1K | 2K | 3K | 4K | 1K | 2K | 3K | 4K |
| Transformer | 8.1 | 4.9 | 2.3 | 1.4 | 0.85 | 2.65 | 5.51 | 9.48 |
| Local Attention | 9.2 (1.1x) | 8.4 (1.7x) | 7.4 (3.2x) | 7.4 (5.3x) | 0.42 | 0.76 | 1.06 | 1.37 |
| Linformer | 9.3 (1.2x) | 9.1 (1.9x) | 8.5 (3.7x) | 7.7 (5.5x) | **0.37** | **0.55** | 0.99 | **0.99** |
| Reformer | 4.4 (0.5x) | 2.2 (0.4x) | 1.5 (0.7x) | 1.1 (0.8x) | 0.48 | 0.99 | 1.53 | 2.28 |
| Sinkhorn Trans | 9.1 (1.1x) | 7.9 (1.6x) | 6.6 (2.9x) | 5.3 (3.8x) | 0.47 | 0.83 | 1.13 | 1.48 |
| Synthesizer | 8.7 (1.1x) | 5.7 (1.2x) | 6.6 (2.9x) | 1.9 (1.4x) | 0.65 | 1.98 | 4.09 | 6.99 |
| BigBird | 7.4 (0.9x) | 3.9 (0.8x) | 2.7 (1.2x) | 1.5 (1.1x) | 0.77 | 1.49 | 2.18 | 2.88 |
| Linear Trans. | 9.1 (1.1x) | 9.3 (1.9x) | 8.6 (3.7x) | 7.8 (5.6x) | **0.37** | 0.57 | **0.80** | 1.03 |
| Performer | **9.5** (1.2x) | **9.4** (1.9x) | **8.7** (3.8x) | **8.0** (5.7x) | **0.37** | 0.59 | 0.82 | 1.06 |

Figure 19: **Upper Table:** Results on Long-Range Arena benchmark. Best model is in boldface and second best is underlined. **Lower Table:** Benchmark results of all X-former models with a consistent batch size of 32 across all models. The authors report relative speed increase/decrease in comparison with the vanilla Transformer in brackets besides the steps per second. Memory usage refers to per device memory usage across each TPU device. Benchmarks are run on 4x4 TPU-v3 chips. **Right Fig:** Performance (y-axis), speed (x-axis), and memory footprint (size of the circles) of different models.

# E COMPUTATION COSTS - EXTENDED RESULTS

In this subsection, we empirically measure computational costs in terms wall clock time on forward and backward passes for three scenarios in Fig. 20:

1. Performer, with varying number of layers. We show that our method can scale up to (but not necessarily limited to) even 20 layers.

2. Attention time complexities when comparing standard attention (from Transformer) and FAVOR (from Performer). Note that the maximum memory size here is not reflective of the maximum memory size in an actual model (shown below), as this benchmark requires computing explicit tensors (causing memory increases) in Jax, while a model does not.

3. Time complexities when comparing the Transformer and Performer models. "X" (OPT) denotes the maximum possible speedup achievable, when attention simply returns the **V**-vector, showing that the Performer is nearly optimal. We see that the maximum possible power of 2 length allowed on a V100 GPU (16GB) is $2^{15} = 32768$ using regular dimensions.

Since some of the computational bottleneck in the Transformer may originate from the extra feed-forward layers (Kitaev et al., 2020), we also benchmark the "Small" version, i.e. $(n_{heads}, n_{layers}, d_{ff}, d) = (1, 6, 64, 64)$ as well, when the attention component is the dominant source of computation and memory. We remind the reader that the "Regular" version consists of $(n_{heads}, n_{layers}, d_{ff}, d) = (8, 6, 2048, 512)$.

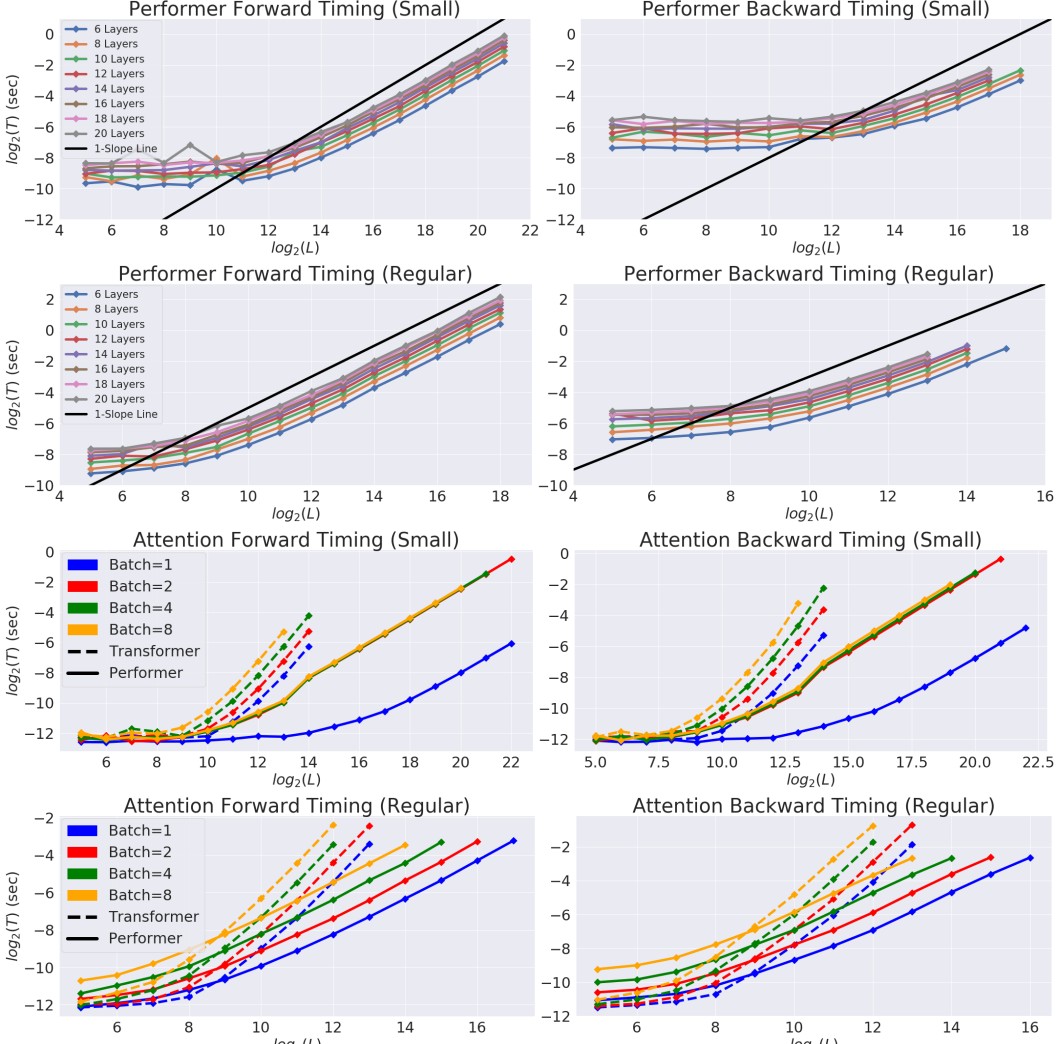

Figure 20: Captions (1) and (2) for each 2x2 subfigure mentioned above.

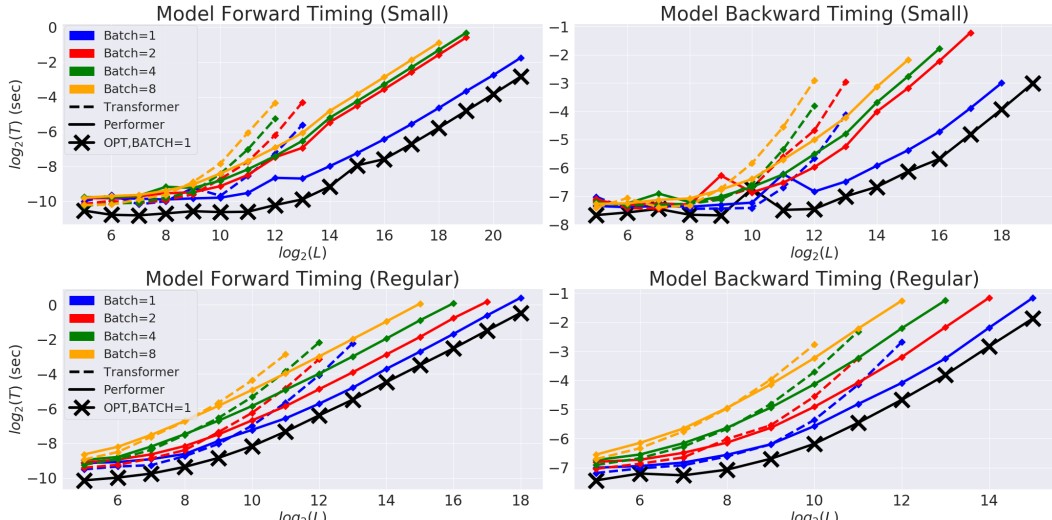

Figure 21: Caption (3) for this 2x2 subfigure mentioned above.

# F    THEORETICAL RESULTS

We provide here the proofs of all theoretical results presented in the paper.

## F.1    PROOF OF LEMMA 1

*Proof.* We first deduce that for any $\boldsymbol{a}, \boldsymbol{b} \in \mathbb{R}^d$

$$\text{SM}(\mathbf{x}, \mathbf{y}) = \exp(\boldsymbol{x}^\top \boldsymbol{y}) = \exp(-\|\boldsymbol{x}\|^2/2) \cdot \exp(\|\boldsymbol{x} + \boldsymbol{y}\|^2/2) \cdot \exp(-\|\boldsymbol{y}\|^2/2).$$

Next, let $\boldsymbol{w} \in \mathbb{R}^d$. We use the fact that

$$(2\pi)^{-d/2} \int \exp(-\|\boldsymbol{w} - \boldsymbol{c}\|_2^2/2) d\boldsymbol{w} = 1$$

for any $\boldsymbol{c} \in \mathbb{R}^d$ and derive:

$$\exp(\|\boldsymbol{x} + \boldsymbol{y}\|^2/2) = (2\pi)^{-d/2} \exp(\|\boldsymbol{x} + \boldsymbol{y}\|^2/2) \int \exp(-\|\boldsymbol{w} - (\boldsymbol{x} + \boldsymbol{y})\|^2/2) d\boldsymbol{w}$$

$$= (2\pi)^{-d/2} \int \exp(-\|\boldsymbol{w}\|^2/2 + \boldsymbol{w}^\top(\boldsymbol{x} + \boldsymbol{y}) - \|\boldsymbol{x} + \boldsymbol{y}\|^2/2 + \|\boldsymbol{x} + \boldsymbol{y}\|^2/2) d\boldsymbol{w}$$

$$= (2\pi)^{-d/2} \int \exp(-\|\boldsymbol{w}\|^2/2 + \boldsymbol{w}^\top(\boldsymbol{x} + \boldsymbol{y})) d\boldsymbol{w}$$

$$= (2\pi)^{-d/2} \int \exp(-\|\boldsymbol{w}\|^2/2) \cdot \exp(\boldsymbol{w}^\top \boldsymbol{x}) \cdot \exp(\boldsymbol{w}^\top \boldsymbol{y}) d\boldsymbol{w}$$

$$= \mathbb{E}_{\omega \sim \mathcal{N}(\mathbf{0}_d, \mathbf{I}_d)}[\exp(\omega^\top \boldsymbol{x}) \cdot \exp(\omega^\top \boldsymbol{y})].$$

That completes the proof of the first part of the lemma. An identity involving hyperbolic cosine function is implied by the fact that for every $\mathbf{u} \in \mathbb{R}^d$ and $\omega \sim \mathcal{N}(0, \mathbf{I}_d)$ the following is true:

$$\mathbb{E}[\exp(\omega^\top \mathbf{u})] = \sum_{i=0}^{\infty} \frac{\mathbb{E}[(\omega^\top \mathbf{u})^{2i}]}{(2i)!} = \frac{1}{2} \sum_{i=0}^{\infty} \frac{\mathbb{E}[(\omega^\top \mathbf{u})^{2i}] + \mathbb{E}[(-\omega^\top \mathbf{u})^{2i}]}{(2i)!}. \tag{12}$$

The cancellation of the odd moments $\mathbb{E}[(\omega^\top \mathbf{u})^{2i+1}]$ follows directly from the fact that $\omega$ is taken from the isotropic distribution (i.e. distribution with pdf function constant on each sphere). That completes the proof. $\qquad\square$

## F.2    PROOF OF LEMMA 2

*Proof.* Denote: $\mathbf{z} = \mathbf{x} + \mathbf{y}$ and $\Delta = \mathbf{x} - \mathbf{y}$. Note that by using standard trigonometric identities (and the fact that the variance of the sum of independent random variables is the sum of variances of those random variables), we can get the following for $\omega \sim \mathcal{N}(0, \mathbf{I}_d)$:

$$\text{MSE}(\widehat{\text{SM}}_m^{\text{trig}}(\mathbf{x}, \mathbf{y})) = \frac{1}{m} \exp(\|\mathbf{x}\|^2 + \|\mathbf{y}\|^2) \text{Var}(\cos(\omega^\top \Delta)). \tag{13}$$

Using the fact that (see: Lemma 1 in (Yu et al., 2016); note that in that lemma they use notation: $z$ for what we denote as: $\|\Delta\|$):

$$\text{Var}(\cos(\omega^\top \Delta)) = \frac{1}{2}(1 - \exp(-\|\Delta\|^2))^2, \tag{14}$$

we obtain:

$$\text{MSE}(\widehat{\text{SM}}_m^{\text{trig}}(\mathbf{x}, \mathbf{y})) = \frac{1}{2m} \exp(\|\mathbf{x}\|^2 + \|\mathbf{y}\|^2)(1 - \exp(-\|\Delta\|^2))^2 =$$

$$\frac{1}{2m} \exp(\|\mathbf{z}\|^2) \text{SM}^{-2}(\mathbf{x}, \mathbf{y})(1 - \exp(-\|\Delta\|^2))^2, \tag{15}$$

which completes the first part of the proof. To obtain the formula for: $\text{MSE}(\widehat{\text{SM}}_m^+(\mathbf{x}, \mathbf{y}))$ notice first that:

$$\mathbb{E}_{\omega \sim \mathcal{N}(0, \mathbf{I}_d)}[\exp(\omega^\top \mathbf{z})] = \exp(\frac{\|\mathbf{z}\|^2}{2}). \tag{16}$$

The above immediately follows from the fact that positive random feature maps provide unbiased estimation of the softmax-kernel, thus the following is true:

$$\mathrm{SM}(\mathbf{x}, \mathbf{y}) = \exp(-\frac{\|\mathbf{x}\|^2 + \|\mathbf{y}\|^2}{2})\mathbb{E}_{\omega \sim \mathcal{N}(0, \mathbf{I}_d)}[\exp(\omega^\top \mathbf{z})]. \tag{17}$$

Therefore we obtain:

$$\mathrm{MSE}(\widehat{\mathrm{SM}}_m^+(\mathbf{x}, \mathbf{y})) = \frac{1}{m}\exp(-(\|\mathbf{x}\|^2 + \|\mathbf{y}\|^2))\mathrm{Var}(\exp(\omega^\top \mathbf{z})) =$$

$$\frac{1}{m}\exp(-(\|\mathbf{x}\|^2 + \|\mathbf{y}\|^2))\left(\mathbb{E}[\exp(2\omega^\top \mathbf{z})] - (\mathbb{E}[\exp(\omega^\top \mathbf{z})])^2\right) = \tag{18}$$

$$\frac{1}{m}\exp(-(\|\mathbf{x}\|^2 + \|\mathbf{y}\|^2))(\exp(2\|\mathbf{z}\|^2) - \exp(\|\mathbf{z}\|^2)),$$

where the last equality follows from Equation 16. Therefore we have:

$$\mathrm{MSE}(\widehat{\mathrm{SM}}_m^+(\mathbf{x}, \mathbf{y})) = \frac{1}{m}\exp(-(\|\mathbf{x}\|^2 + \|\mathbf{y}\|^2))\exp(\|\mathbf{z}\|^2)(\exp(\|\mathbf{z}\|^2) - 1) =$$
$$\frac{1}{m}\exp(\|\mathbf{z}\|^2)\mathrm{SM}^2(\mathbf{x}, \mathbf{y})(1 - \exp(-\|\mathbf{z}\|^2)). \tag{19}$$

Finally,

$$\mathrm{MSE}(\widehat{\mathrm{SM}}_m^{\mathrm{hyp}+}(\mathbf{x}, \mathbf{y})) = \frac{1}{4m}\exp(-\frac{\|\mathbf{x}\|^2 + \|\mathbf{y}\|^2}{2})^2(\mathrm{Var}(\exp(\omega^\top \mathbf{z})) + \mathrm{Var}(\exp(-\omega^\top \mathbf{z}))+$$

$$2\mathrm{Cov}(\exp(\omega^\top \mathbf{z})), \exp(-\omega^\top \mathbf{z})))) = \frac{1}{4m}\exp(-\frac{\|\mathbf{x}\|^2 + \|\mathbf{y}\|^2}{2})^2(2\mathrm{Var}(\exp(\omega^\top \mathbf{z}))+$$

$$2\mathrm{Cov}(\exp(\omega^\top \mathbf{z})), \exp(-\omega^\top \mathbf{z}))))) = \frac{1}{2m}\exp(-(\|\mathbf{x}\|^2 + \|\mathbf{y}\|^2))$$

$$(\mathrm{Var}(\exp(\omega^\top \mathbf{z})) + 1 - (\mathbb{E}[\exp(\omega^\top \mathbf{z})])^2) = \frac{1}{2m}\exp(-(\|\mathbf{x}\|^2 + \|\mathbf{y}\|^2))$$

$$(\exp(2\|\mathbf{z}\|^2) - \exp(\|\mathbf{z}\|^2) + 1 - \exp(\|\mathbf{z}\|^2)) = \frac{1}{2m}\exp(-(\|\mathbf{x}\|^2 + \|\mathbf{y}\|^2))(\exp(\|\mathbf{z}\|^2) - 1)^2$$

$$= \frac{1}{2}(1 - \exp(-\|\mathbf{z}\|^2))\mathrm{MSE}(\widehat{\mathrm{SM}}_m^+(\mathbf{x}, \mathbf{y})). \tag{20}$$

In the chain of equalities above we used the fact that random variables $\exp(\omega^\top \mathbf{z})$ and $\exp(-\omega^\top \mathbf{z})$ have the same distribution. This is true since $\omega$ and $-\omega$ have the same distribution ($\omega$ is Gaussian). That completes the proof. $\qquad \square$

### F.3   PROOF OF THEOREM 1

*Proof.* Let $\mathbf{x}, \mathbf{y} \in \mathbb{R}^d$ be respectively a query/key. Note that from the definition of $\mathrm{SMREG}(\mathbf{x}, \mathbf{y})$ we have for $\mathbf{z} = \mathbf{x} + \mathbf{y}$:

$$\mathrm{SMREG}(\mathbf{x}, \mathbf{y}) = \exp(-\frac{\|\mathbf{x}\|^2 + \|\mathbf{y}\|^2}{2})\sum_{k=0}^{\infty}\frac{1}{(2k)!}\|\mathbf{z}\|^{2k}d^k\mathbb{E}_{\omega \sim \mathcal{N}(0, \mathbf{I}_d)}[(\frac{\omega}{\|\omega\|_2}\mathbf{e}_1)^{2k}], \tag{21}$$

where $\mathbf{e}_1 \overset{\mathrm{def}}{=} (1, 0, ..., 0)^\top \in \mathbb{R}^d$. To obtain the above we used the fact that $\mathcal{N}(0, \mathbf{I}_d)$ is isotropic (that in particular implies zeroing of the even terms in the Taylor expansion).

Let us denote: $A(k, d) \overset{\mathrm{def}}{=} \mathbb{E}_{\omega \sim \mathcal{N}(0, \mathbf{I}_d)}[(\frac{\omega}{\|\omega\|_2}\mathbf{e}_1)^{2k}]$. It turns out that:

$$A(2k, d) = \frac{(2k - 1)!!}{(d + 2k - 2)(d + 2k - 4) \cdot ... \cdot d}. \tag{22}$$

The proof of that fact can be found in the supplement of (Choromanski et al., 2018b), yet we provide it below for completeness and the convenience of the Reader:

**Lemma 3.** *Expression $A(2k, d)$ satisfies the following for $k \in \mathbb{N}$ :*

$$A(2k, d) = \frac{(2k-1)!!}{(d+2k-2)(d+2k-4) \cdot \ldots \cdot d}. \tag{23}$$

*Proof.* Note first that for $d \geq 2$ the density function $p_d(\theta)$ of the angle between a vector $\mathbf{r} \in \mathbb{R}^d$ chosen uniformly at random from the unit sphere and $\mathbf{e}_1$ is given by the following formula:

$$p_d(\theta) = \frac{\sin^{d-2}(\theta)}{\int_0^\pi \sin^{d-2(\theta)} d\theta}. \tag{24}$$

Let us denote: $F(k, d) \overset{\text{def}}{=} \int_0^\pi \cos^k(\theta) \sin^d(\theta) d\theta$. Using partial integration, we get:

$$\int_0^\pi \cos^k(\theta) \sin^d(\theta) d\theta = \int_0^\pi \cos^{k-1}(\theta) \sin^d(\theta)(\sin(\theta))' d\theta =$$

$$\cos^{k-1}(\theta) \sin^{d+1}(\theta)|_0^\pi - \int_0^\pi \sin(\theta)((k-1)\cos^{k-2}(\theta)(-\sin(\theta))\sin^d(\theta)+ \tag{25}$$

$$d\cos^k(\theta)\sin^{d-1}(\theta))d\theta.$$

Thus we conclude that: $F(k, d) = \frac{k-1}{d+1} F(k-2, d+2)$. Therefore we have:

$$F(2k, d) = \frac{(2k-1)!!}{(d+1)(d+3) \cdot \ldots \cdot (d+2k-1)} \int_0^\pi \sin^{d+2k}(\theta) d\theta. \tag{26}$$

We again conduct partial integration and get:

$$\int_0^\pi \sin^d(\theta) d\theta = -\frac{1}{d} \sin^{d-1}(\theta) \cos(\theta)|_0^\pi +$$

$$\frac{d-1}{d} \int_0^\pi \sin^{d-2}(\theta) d\theta = \frac{d-1}{d} \int_0^\pi \sin^{d-2}(\theta) d\theta. \tag{27}$$

Therefore we conclude that:

$$A(2k, d) = \frac{1}{\frac{d-3}{d-2}\frac{d-5}{d-4} \cdot \ldots} \frac{(2k-1)!!}{(d-1)(d+1) \cdot \ldots \cdot (d+2k-3)} \frac{d+2k-3}{d+2k-2}\frac{d+2k-5}{d+2k-4} \cdot \ldots =$$

$$\frac{(2k-1)!!}{(d+2k-2)(d+2k-4) \cdot \ldots \cdot d}, \tag{28}$$

which completes the proof. $\qquad\square$

Applying the above lemma, we get:

$$\text{SMREG}(\mathbf{x}, \mathbf{y}) = \exp(-\frac{\|\mathbf{x}\|^2 + \|\mathbf{y}\|^2}{2}) \sum_{k=0}^\infty \frac{1}{(2k)!} \|\mathbf{z}\|^{2k} d^k \frac{(2k-1)!!}{(d+2k-2)(d+2k-4) \cdot \ldots \cdot d}$$

$$= \exp(-\frac{\|\mathbf{x}\|^2 + \|\mathbf{y}\|^2}{2}) \sum_{k=0}^\infty \frac{w^k}{k!} f(k, d), \tag{29}$$

where $w = \frac{\|\mathbf{z}\|^2}{2}$ and $f(k, d) = \frac{d^k}{(d+2k-2)(d+2k-4) \cdot \ldots \cdot d}$.

Thus we obtain:

$$\frac{\text{SMREG}(\mathbf{x}, \mathbf{y})}{\text{SM}(\mathbf{x}, \mathbf{y})} = e^{-w} \sum_{k=0}^\infty \frac{w^k}{k!} f(k, d). \tag{30}$$

Note first that for $k \geq 1$ we have: $f(k, d) \leq 1$, thus:

$$\text{SMREG}(\mathbf{x}, \mathbf{y}) \leq \text{SM}(\mathbf{x}, \mathbf{y}). \tag{31}$$

We also have for $l = d^{\frac{1}{3}}$:

$$\frac{\text{SMREG}(\mathbf{x}, \mathbf{y})}{\text{SM}(\mathbf{x}, \mathbf{y})} = e^{-w} \sum_{k=0}^{l} \frac{w^k}{k!} f(k, d) + e^{-w} \sum_{k=l+1}^{\infty} \frac{w^k}{k!} f(k, d) \geq$$

$$f(l, d) e^{-w} \sum_{k=0}^{l} \frac{w^k}{k!} + e^{-w} \sum_{k=l+1}^{\infty} \frac{w^k}{k!} f(k, d) \geq f(l, d)(1 - e^{-w} \sum_{k=l+1}^{\infty} \frac{w^k}{k!}) = \tag{32}$$

$$f(l, d)(1 - \mathbb{P}[\text{Po}(w) > l]),$$

where $\text{Po}(w)$ stands for the random variable of Poisson distribution with parameter $w$. Therefore we get for $t = \ln(\frac{l}{w})$:

$$\frac{\text{SMREG}(\mathbf{x}, \mathbf{y})}{\text{SM}(\mathbf{x}, \mathbf{y})} \geq (1 - \frac{2l - 2}{d})^l (1 - \mathbb{P}[\text{Po}(w) > l]) \geq$$

$$\exp(l \ln(1 - \frac{2l - 2}{d}))(1 - \mathbb{P}[t\text{Po}(w) \geq tl]) =$$

$$\exp\left( l \sum_{i=1}^{\infty} (-1)^i \frac{(\frac{2l-2}{d})^i}{i} \right) (1 - \mathbb{P}[\exp(t\text{Po}(w) - tl) \geq 1]) \geq \tag{33}$$

$$\exp(-\frac{2}{d^{\frac{1}{3}}} + o(\frac{1}{d^{\frac{1}{3}}}))(1 - \exp(-tl)\mathbb{E}[\exp(t\text{Po}(w))]) =$$

$$\exp(-\frac{2}{d^{\frac{1}{3}}} + o(\frac{1}{d^{\frac{1}{3}}}))(1 - \exp(-w - l(t - 1))),$$

where the last equality is implied by the formula for the Laplace Transform for the Poisson random variable:

$$\mathbb{E}[\exp(t\text{Po}(w))] = \exp(w(\exp(t) - 1)). \tag{34}$$

Notice that: $w = \frac{\|\mathbf{z}\|^2}{2} = \frac{\ln(\text{SM}(\mathbf{x},\mathbf{x})) + \ln(\text{SM}(\mathbf{y},\mathbf{y})) + 2\ln(\text{SM}(\mathbf{x},\mathbf{y}))}{2} \leq 2\ln(C)$. We conclude that:

$$\frac{\text{SMREG}(\mathbf{x}, \mathbf{y})}{\text{SM}(\mathbf{x}, \mathbf{y})} \geq (1 - \frac{2}{d^{\frac{1}{3}}} + o(\frac{1}{d^{\frac{1}{3}}}))(1 - C^{-2}(\frac{d^{\frac{1}{3}}}{2e \cdot \ln(C)})^{-d^{\frac{1}{3}}}) = 1 - \frac{2}{d^{\frac{1}{3}}} + o(\frac{1}{d^{\frac{1}{3}}}). \tag{35}$$

That completes the proof. □

### F.4 Proofs of Theorem 2, Theorem 3 & Beautiful Functions

We will provide here much more general theoretical results which will imply Theorem 3 and Theorem 2. We need the following definition:

**Definition 1.** *We say that function $F : \mathbb{R}^n \to \mathbb{R}$ is beautiful if $F$ can be expressed as:*

$$F_{\Omega,g}(\mathbf{z}) = \mathbb{E}_{\omega \sim \Omega}[g(\omega^\top \mathbf{z})], \tag{36}$$

*for a probabilistic isotropic distribution $\Omega$, and where $g : \mathbb{R} \to \mathbb{R}$ is an entire function with non-negative power-series coefficients (i.e. $g(x) = \sum_{i=0}^{\infty} a_i x^i$ for every $x \in \mathbb{R}$ and with $a_i \geq 0$ for $i = 0, 1, ...$). In the formula above we assume that the expectation on the RHS exists.*

Interestingly, beautiful functions can be used to define softmax and consequently, Gaussian kernels (both standard and regularized), leading to our PRF mechanism presented in the main body of the paper, as we explain below.

**Remark 1.** *If one takes $\Omega = \mathcal{N}(0, \mathbf{I}_d)$ (note that $\mathcal{N}(0, \mathbf{I}_d)$ is isotropic) and $g : x \to \exp(x)$ (such $g$ is clearly entire with nonnegative power-series coefficient) then the following is true for $\mathbf{z} = \mathbf{x} + \mathbf{y}$:*

$$\text{SM}(\mathbf{x}, \mathbf{y}) = \exp(-\frac{\|\mathbf{x}\|^2 + \|\mathbf{y}\|^2}{2})F_{\Omega,g}(\mathbf{z}). \tag{37}$$

*Similarly:* $\text{SMREG}(\mathbf{x}, \mathbf{y}) = \exp(-\frac{\|\mathbf{x}\|^2 + \|\mathbf{y}\|^2}{2})F_{\Omega_{\text{reg}},g}(\mathbf{z})$, *where $\Omega_{\text{reg}}$ stands for the distribution corresponding to Haar measure on the sphere of radius $\sqrt{d}$ (which is clearly isotropic). Therefore general concentration results for Monte Carlo estimators of beautiful functions immediately imply corresponding results for the (standard and regularized) softmax (and thus also Gaussian) kernel.*

We will consider two estimators of the beautiful functions from Definition 1 that directly lead (through Remark 1) to: PRF-based approximation of the softmax-kernel and its enhanced version with orthogonal features. Standard Monte Carlo estimator samples independently $\omega_1^{\mathrm{iid}}, ..., \omega_m^{\mathrm{iid}} \overset{\mathrm{iid}}{\sim} \Omega$, where $m$ stands for the number of samples and then computes:

$$\widehat{F}_m^{\mathrm{iid}}(\mathbf{z}) \overset{\mathrm{def}}{=} \frac{1}{m} \sum_{i=1}^{m} g((\omega_i^{\mathrm{iid}})^\top \mathbf{z}). \tag{38}$$

Orthogonal Monte Carlo estimator samples $\omega_1^{\mathrm{ort}}, ..., \omega_m^{\mathrm{ort}}$ ($m \leq d$) in such a way that marginally we have: $\omega_i^{\mathrm{ort}} \sim \Omega$, but $(\omega_i^{\mathrm{ort}})^\top \omega_j^{\mathrm{ort}} = 0$ for $i \neq j$ (such an orthogonal ensemble can be always created if $\Omega$ is isotropic, as we already mentioned in the main body of the paper). We define:

$$\widehat{F}_m^{\mathrm{ort}}(\mathbf{z}) \overset{\mathrm{def}}{=} \frac{1}{m} \sum_{i=1}^{m} g((\omega_i^{\mathrm{ort}})^\top \mathbf{z}). \tag{39}$$

### F.4.1 ORTHOGONALITY UNIVERSALLY IMPROVES CONCENTRATION

Denote by $M_Z(\theta) = \mathbb{E}[e^{\theta Z}]$ a moment generating function of the random variable $Z$. Note first that estimators of beautiful functions based on standard Monte Carlo procedure using independent vectors $\omega_i^{\mathrm{iid}}$ guarantee strong concentration bounds since independent $\omega_i$s provide a way to obtain exponentially small upper bounds on failure probabilities through moment generating functions. We summarize this classic observation which is a standard application of Markov's Inequality below.

**Lemma 4.** *Consider an estimator $\widehat{F}_m^{\mathrm{iid}}(\mathbf{z})$ of the beautiful function $F$ evaluated at $\mathbf{z}$. Then the following holds for any $a > F(\mathbf{z})$:*

$$\mathbb{P}[\widehat{F}_m^{\mathrm{iid}}(\mathbf{z}) > a] \leq e^{-m\mathcal{L}_X(a)}, \tag{40}$$

*where $X = g(\mathbf{w}^\top \mathbf{z})$, $\mathbf{w} \sim \mathcal{D}$ and $\mathcal{L}_Z$ stands for a Legendre Transform of the random variable $Z$ defined as: $\mathcal{L}_Z(a) = \sup_{\theta > 0} \log(\frac{e^{\theta a}}{M_Z(\theta)})$. Furthermore, $\mathcal{L}_X(a) > 0$.*

The above result provides us with exponentially small (in Legendre Transform) upper bounds on tail probabilities for the standard estimator. Below we provide our two main theoretical results.

**Theorem 5** (orthogonality provides smaller tails)**.** *If $F_{\Omega,g}$ is a beautiful function then the following holds for $m \leq d$, $X$ as in Lemma 4 and any $a > F(\mathbf{z})$:*

$$\mathbb{P}[\widehat{F}_m^{\mathrm{ort}}(\mathbf{z})) > a] \leq \frac{d}{d+2} e^{-m\mathcal{L}_X(a)}. \tag{41}$$

This result shows that features obtained from the ensembles of pairwise orthogonal random vectors provide exponentially small bounds on tail probabilities and that these bounds are strictly better than for estimators using unstructured features. Furthermore, the result is **universal**, i.e. holds for any dimensionality $d$, not just asymptotically for $d$ large enough.

We also obtain similar result regarding mean squared errors (MSEs) of the considered estimators:

**Theorem 6.** *If $F_{\Omega,g}$ is a beautiful function then the following holds for $m \leq d$:*

$$\mathrm{MSE}(\widehat{F}_m^{\mathrm{ort}}(\mathbf{z})) \leq \mathrm{MSE}(\widehat{F}_m^{\mathrm{iid}}(\mathbf{z})) - (1 - \frac{1}{m})\frac{2}{d+2} F_{\Omega,g}^2(\mathbf{z}). \tag{42}$$

As before, an orthogonal estimator leads to better concentration results and as before, this is the case for any $d > 0$, not only asymptotically for large enough $d$.

**Note that from what we have said above, Theorem 2 and Theorem 3 follow immediately from Theorem 6 and Theorem 5 respectively.**

Thus in the remainder of this section we will prove Theorem 6 and Theorem 5.

### F.4.2 PROOF OF THEOREM 5

*Proof.* Note that by the analogous application of Markov's Inequality as in Lemma 4, we get:

$$\mathbb{P}[\widehat{F}_m^{\text{ort}}(\mathbf{z})) > a] \leq \frac{\mathbb{E}[e^{\theta(X_1^{\text{ort}}+...+X_m^{\text{ort}})}]}{e^{\theta m a}}, \tag{43}$$

where we have: $X_i^{\text{ort}} = g((\omega_i^{\text{ort}})^\top \mathbf{z})$. We see that it suffices to show that for any $\theta > 0$ the following holds: $\mathbb{E}[e^{\theta(X_1^{\text{ort}}+...+X_m^{\text{ort}})}] < \mathbb{E}[e^{\theta(X_1^{\text{iid}}+...+X_m^{\text{iid}})}]$. We have:

$$\mathbb{E}[e^{\theta(X_1^{\text{ort}}+...+X_m^{\text{ort}})}] = \mathbb{E}[\sum_{j=0}^{\infty} \frac{(\theta \sum_{i=1}^m X_i^{\text{ort}})^j}{j!}] = \mathbb{E}[\sum_{j=0}^{\infty} \frac{\theta^j}{j!}(\sum_{i=1}^m X_i^{\text{ort}})^j] =$$

$$\sum_{j=0}^{\infty} \frac{\theta^j}{j!} \mathbb{E}[(\sum_{i=1}^m X_i^{\text{ort}})^j] = \sum_{j=0}^{\infty} \frac{\theta^j}{j!} \mathbb{E}[\sum_{(j_1,...,j_m)\in\mathcal{S}_j} c(j_1,...,j_m)(X_1^{\text{ort}})^{j_1} \cdot ... \cdot (X_m^{\text{ort}})^{j_m}], \tag{44}$$

where $\mathcal{S}_j = \{(j_1,...,j_m) \in \mathbb{N} \times ... \times \mathbb{N} : j_1,...,j_m \geq 0, j_1 + ... + j_m = j\}$ and for some positive constants $c(j_1,...,j_m)$.

Thus we have:

$$\mathbb{E}[e^{\theta(X_1^{\text{ort}}+...+X_m^{\text{ort}})}] = \sum_{j=0}^{\infty} \frac{\theta^j}{j!} \sum_{(j_1,...,j_m)\in\mathcal{S}_j} c(j_1,...,j_m)\mathbb{E}[(X_1^{\text{ort}})^{j_1} \cdot ... \cdot (X_m^{\text{ort}})^{j_m}]. \tag{45}$$

Similarly, we get:

$$\mathbb{E}[e^{\theta(X_1^{\text{iid}}+...+X_m^{\text{iid}})}] = \sum_{j=0}^{\infty} \frac{\theta^j}{j!} \sum_{(j_1,...,j_m)\in\mathcal{S}_j} c(j_1,...,j_m)\mathbb{E}[(X_1^{\text{iid}})^{j_1} \cdot ... \cdot (X_m^{\text{iid}})^{j_m}]. \tag{46}$$

Therefore we get:

$$\Delta = \mathbb{E}[e^{\theta(X_1^{\text{iid}}+...+X_m^{\text{iid}})}] - \mathbb{E}[e^{\theta(X_1^{\text{ort}}+...+X_m^{\text{ort}})}]$$

$$= \sum_{j=0}^{\infty} \frac{\theta^j}{j!} \sum_{(j_1,...,j_m)\in\mathcal{S}_j} c(j_1,...,j_m) \left(\mathbb{E}[(X_1^{\text{iid}})^{j_1} \cdot ... \cdot (X_m^{\text{iid}})^{j_m}] - \mathbb{E}[(X_1^{\text{ort}})^{j_1} \cdot ... \cdot (X_m^{\text{ort}})^{j_m}]\right) \tag{47}$$

Note first that using the fact that $f$ is entire, we can rewrite each $X_i^{\text{ort}}$ as:

$$X_i^{\text{ort}} = \sum_{s=0}^{\infty} a_s((\omega_i^{\text{ort}})^\top \mathbf{z})^s, \tag{48}$$

where $f(x) = \sum_{s=0}^{\infty} a_s x^s$ and $a_0, a_1, ... \geq 0$. Similarly,

$$X_i^{\text{iid}} = \sum_{s=0}^{\infty} a_s((\omega_i^{\text{iid}})^\top \mathbf{z})^s. \tag{49}$$

By plugging in the above formulae for $X_i^{\text{ort}}$ and $X_i^{\text{iid}}$ int the formula for $\Delta$ and expanding power-expressions, we obtain:

$$\Delta = \sum_{j=0}^{\infty} \frac{\theta^j}{j!} \sum_{(j_1,...,j_m)\in\mathcal{S}_j} c(j_1,...,j_m) \sum_{(d_1,...,d_m)\in\mathcal{D}(j_1,...,j_m)} \widehat{\Delta}(d_1,...,d_m), \tag{50}$$

for some ordered subsets of indices (with potentially repeating entries) $\mathcal{D}(j_1,...,j_m)$ (exact formula for those can be given but we do not need it to complete the proof and since it is technical, it would unnecessarily complicate the proof so we skip it) and $\widehat{\Delta}(d_1,...,d_m)$ defined as:

$$\widehat{\Delta}(d_1,...,d_m) = \mathbb{E}[((\omega_1^{\text{iid}})^\top \mathbf{z})^{d_1} \cdot ... \cdot ((\omega_m^{\text{iid}})^\top \mathbf{z})^{d_m}] - \mathbb{E}[((\omega_1^{\text{ort}})^\top \mathbf{z})^{d_1} \cdot ... \cdot ((\omega_m^{\text{ort}})^\top \mathbf{z})^{d_m}]. \tag{51}$$

Our next goal is to re-write the formula for $\widehat{\Delta}(d_1, ..., d_m)$. Denote:

$$Y = ((\omega_1^{\text{ort}})^\top \mathbf{z})^{d_1} \cdot ... \cdot ((\omega_m^{\text{ort}})^\top \mathbf{z})^{d_m}. \tag{52}$$

Observe that $Y$ has the same distribution as $Y'$ defined as:

$$Y' = (\mathbf{e}_1^\top \frac{\mathbf{g}}{\|\mathbf{g}\|_2} \|\mathbf{z}\|_2)^{d_1} \cdot ... \cdot (\mathbf{e}_m^\top \frac{\mathbf{g}}{\|\mathbf{g}\|_2} \|\mathbf{z}\|_2)^{d_m} \cdot (\|\omega_1^{\text{ort}}\|_2)^{d_1} \cdot ... \cdot (\|\omega_m^{\text{ort}}\|_2)^{d_m}, \tag{53}$$

where $\mathbf{g}$ is a Gaussian vector taken from the $\mathcal{N}(0, \mathbf{I}_d)$ distribution, independently from: $\|\omega_1^{\text{ort}}\|_2, ..., \|\omega_m^{\text{ort}}\|_2$.

This comes from the fact that for a fixed $\mathbf{z}$ one can think about the set: $\frac{\omega_1^{\text{ort}}}{\|\omega_1^{\text{ort}}\|_2}, ..., \frac{\omega_m^{\text{ort}}}{\|\omega_m^{\text{ort}}\|_2}$ as a random rotation of the system of $m$ canonical basis vectors: $\mathbf{e}_1, ..., \mathbf{e}_m$. Thus instead of applying a random rotation to: $\mathbf{e}_1, ..., \mathbf{e}_m$, one can equivalently randomly rotate vector $\mathbf{z}$. Randomly rotated vector $\mathbf{z}$ has the same distribution as: $\frac{\mathbf{g}}{\|\mathbf{g}\|_2} \|\mathbf{z}\|_2$.

Now note that lengths of vectors $\omega_1^{\text{ort}}, ..., \omega_m^{\text{ort}}$ are chosen independently.

Therefore we obtain:

$$\mathbb{E}[((\omega_1^{\text{ort}})^\top \mathbf{z})^{d_1} \cdot ... \cdot ((\omega_m^{\text{ort}})^\top \mathbf{z})^{d_m}] =$$
$$\mathbb{E}[(\|\omega_1^{\text{ort}}\|_2)^{d_1}] \cdot ... \cdot \mathbb{E}[(\|\omega_m^{\text{ort}}\|_2)^{d_m}] \cdot \mathbb{E}[(\mathbf{e}_1^\top \mathbf{v})^{d_1} \cdot ... \cdot (\mathbf{e}_m^\top \mathbf{v})^{d_m}] \|\mathbf{z}\|_2^{d_1 + ... + d_m}, \tag{54}$$

where $\mathbf{v} \sim \frac{\mathbf{g}}{\|\mathbf{g}\|_2}$.

Denote $\mathbf{g} = (g_1, ..., g_d)^\top$. Thus we obtain:

$$\mathbb{E}[((\omega_1^{\text{ort}})^\top \mathbf{z})^{d_1} \cdot ... \cdot ((\omega_m^{\text{ort}})^\top \mathbf{z})^{d_m}] =$$
$$\mathbb{E}[(\|\omega_1^{\text{ort}}\|_2)^{d_1}] \cdot ... \cdot \mathbb{E}[(\|\omega_m^{\text{ort}}\|_2)^{d_m}] \cdot \|\mathbf{z}\|_2^{d_1 + ... + d_m} \mathbb{E}[\frac{g_1^{d_1} \cdot ... \cdot g_m^{d_m}}{\sqrt{g_1^2 + ... + g_d^2}^{d_1 + ... + d_m}}] \tag{55}$$

Now let us focus on the second expression from the formula on $\widehat{\Delta}(d_1, ..., d_m)$. We have:

$$\mathbb{E}[((\omega_1^{\text{iid}})^\top \mathbf{z})^{d_1} \cdot ... \cdot ((\omega_m^{\text{iid}})^\top \mathbf{z})^{d_m}] = \prod_{i=1}^m \mathbb{E}[((\omega_i^{\text{iid}})^\top \mathbf{z})^{d_i}] =$$
$$\mathbb{E}[(\|\omega_1^{\text{iid}}\|_2)^{d_1}] \cdot ... \cdot \mathbb{E}[(\|\omega_m^{\text{iid}}\|_2)^{d_m}] \cdot \|\mathbf{z}\|_2^{d_1 + ... + d_m} \cdot \prod_{i=1}^m \mathbb{E}[\frac{g_i^{d_i}}{\sqrt{g_1^2 + ... + g_d^2}^{d_i}}], \tag{56}$$

where the first equality comes from the fact that different $\omega_i^{\text{iid}}$s are independent and the second one is implied by the analogous analysis to the one conducted above.

We will need the following lemma:

**Lemma 5.** *For every $s \in \mathbb{N}_+$ such that $s \le n$ and every $k_1, ..., k_s \in \mathbb{N}_+$ the following holds:*

$$\mathbb{E}[\frac{g_1^{k_1} \cdot ... \cdot g_s^{k_s}}{\sqrt{g_1^2 + ... + g_d^2}^{k_1 + ... + k_s}}] = \frac{\prod_{i=1}^s \mathbb{E}[g_i^{k_i}]}{\mathbb{E}[\sqrt{g_1^2 + ... + g_d^2}^{k_1 + ... + k_s}]}. \tag{57}$$

*Proof.* Take $\mathbf{r} = \frac{\mathbf{g}}{\|\mathbf{g}\|_2} \|\tilde{\mathbf{g}}\|_2$, where $\tilde{\mathbf{g}}$ is an independent copy of $\mathbf{g}$. Note that $\mathbf{r} \sim \mathbf{g}$. We have:

$$\mathbb{E}[r_1^{k_1}] \cdot ... \cdot \mathbb{E}[r_s^{k_s}] = \mathbb{E}[r_1^{k_1} \cdot ... \cdot r_s^{k_s}] = \mathbb{E}[\frac{g_1^{k_1} \cdot ... \cdot g_s^{k_s}}{\sqrt{g_1^2 + ... + g_d^2}^{k_1 + ... + k_s}}] \cdot \mathbb{E}[\|\tilde{\mathbf{g}}\|_2^{k_1 + ... + k_s}], \tag{58}$$

where the first equality comes from the independence of different elements of $\mathbf{z} = (z_1, ..., z_n)^\top$ and the second equality is implied by the fact that $\tilde{\mathbf{g}}$ is independent from $\mathbf{g}$.

Therefore we have:

$$\mathbb{E}\left[\frac{g_1^{k_1} \cdot ... \cdot g_s^{k_s}}{\sqrt{g_1^2 + ... + g_d^2}^{k_1 + ... + k_s}}\right] = \frac{\mathbb{E}[r_1^{k_1}] \cdot ... \cdot \mathbb{E}[r_s^{k_s}]}{\mathbb{E}[\|\tilde{\mathbf{g}}\|_2^{k_1 + ... + k_s}]}. \tag{59}$$

That completes the proof since $\mathbf{z} \sim \mathbf{g}$ and $\tilde{\mathbf{g}} \sim \mathbf{g}$. $\qquad\square$

Note that by Lemma 5, we can rewrite the right expression from the formula on $\widehat{\Delta}(d_1, ..., d_m)$ as:

$$\mathbb{E}[(\|\omega_1^{\text{ort}}\|_2)^{d_1}] \cdot ... \cdot \mathbb{E}[(\|\omega_m^{\text{ort}}\|_2)^{d_m}] \cdot \|\mathbf{z}\|_2^{d_1 + ... + d_m} \frac{\prod_{i=1}^m \mathbb{E}[g_i^{d_i}]}{\mathbb{E}[\sqrt{g_1^2 + ... + g_d^2}^{d_1 + ... + d_m}]}. \tag{60}$$

The left expression from the formula on $\widehat{\Delta}(d_1, ..., d_m)$ can be rewritten as:

$$L(d_1, ..., d_m) = \mathbb{E}[(\|\omega_1^{\text{iid}}\|_2)^{d_1}] \cdot ... \cdot \mathbb{E}[(\|\omega_m^{\text{iid}}\|_2)^{d_m}] \cdot \|\mathbf{z}\|_2^{d_1 + ... + d_m}$$
$$\frac{\prod_{i=1}^m \mathbb{E}[g_i^{d_i}]}{\mathbb{E}[\sqrt{g_1^2 + ... + g_d^2}^{d_1}] \cdot ... \cdot \mathbb{E}[\sqrt{g_1^2 + ... + g_d^2}^{d_m}]}. \tag{61}$$

Since marginal distributions of $\omega_i^{\text{ort}}$ and $\omega_i^{\text{iid}}$ are the same, we can rewrite $\widehat{\Delta}(d_1, ..., d_n)$ as:

$$\widehat{\Delta}(d_1, ..., d_m) = L(d_1, ..., d_m)(1 - \tau(d_1, ..., d_m)), \tag{62}$$

where $\tau(d_1, ..., d_m)$ is defined as:

$$\tau(d_1, ..., d_m) = \frac{\mathbb{E}[\sqrt{g_1^2 + ... + g_d^2}^{d_1}] \cdot ... \cdot \mathbb{E}[\sqrt{g_1^2 + ... + g_d^2}^{d_m}]}{\mathbb{E}[\sqrt{g_1^2 + ... + g_d^2}^{d_1 + ... + d_m}]} \tag{63}$$

We need now few observations regarding $\widehat{\Delta}(d_1, ..., d_m)$. Note firsr that since odd moments of the Gaussian scalar distribution $\mathcal{N}(0, 1)$ are zero, $\widehat{\Delta}(d_1, ..., d_m)$ is zero if at least of of $d_i$ is odd. Furthermore, $\widehat{\Delta(d_1, ..., d_m)}$ is trivially zero if all but at most one $d_i$ are zero.

With our new notation, $\Delta$ can be rewritten as:

$$\Delta = \sum_{j=0}^\infty \frac{\theta^j}{j!} \sum_{(j_1, ..., j_m) \in \mathcal{S}_j} c(j_1, ..., j_m) \sum_{(d_1, ..., d_m) \in \mathcal{D}(j_1, ..., j_m)} L(d_1, ..., d_m)(1 - \tau(d_1, ..., d_m)), \tag{64}$$

Note also that we have:

$$e^{\theta(X_1^{\text{iid}} + ... + X_m^{\text{iid}})} = \sum_{j=0}^\infty \frac{\theta^j}{j!} \sum_{(j_1, ..., j_m) \in \mathcal{S}_j} c(j_1, ..., j_m) \sum_{(d_1, ..., d_m) \in \mathcal{D}(j_1, ..., j_m)} L(d_1, ..., d_m). \tag{65}$$

Therefore (see: our observations on $\widehat{\Delta}(d_1, ..., d_m)$) to complete the proof it suffices to show that: $\tau(d_1, ..., d_m) \leq \frac{d}{d+2}$ if at least two: $d_i, d_j$ for $i \neq j$ are nonzero and all $d_i$ are even.

**Lemma 6.** *The following holds if for some $i \neq j$ we have: $d_i, d_j > 0$ and all $d_i$ are even:*

$$\tau(d_1, ..., d_m) \leq \frac{d}{d + 2}. \tag{66}$$

*Proof.* Note that $\tau(d_1, ..., d_m)$ can be rewritten as:

$$\tau(d_1, ..., d_m) = \frac{\prod_{i=1}^m \mu_d(d_i)}{\mu_d(\sum_{i=1}^m d_i)}, \tag{67}$$

where $\mu_d(j)$ stands for the $j^{th}$ moment of the $\chi$-distribution with $d$ degrees of freedom. Note that $\mu_d(j) = 2^{\frac{j}{2}} \frac{\Gamma(\frac{d+j}{2})}{\Gamma(\frac{d}{2})}$, where $\Gamma$ is the so-called *Gamma-function*.

Using the fact that: $\Gamma(n) = (n-1)!$ and $\Gamma(n + \frac{1}{2}) = \frac{(2n-1)!!}{2^n} \sqrt{\pi}$ for $n \in \mathbb{N}_+$, it is easy to see that for a fixed $d$, the RHS of the Equality 67 is maximized when $d_i = d_j = 2$ and $d_k = 0$ for some $i \neq j$ and $k \notin \{i, j\}$. Furthermore, straightforward calculations show that in that case the value of the RHS from Equality 67 is $\frac{d}{d+2}$. That completes the proof of the Lemma, and consequently, the proof of the entire Theorem. $\qquad\square$

### F.4.3   PROOF OF THEOREM 6

$\square$

*Proof.* We will use the notation from the proof of Theorem 5. Since both estimators: $\widehat{F}_m^{\mathrm{ort}}(\mathbf{z})$ and $\widehat{F}_m^{\mathrm{iid}}(\mathbf{z})$ are unbiased, we have: $\mathrm{MSE}(\widehat{F}_m^{\mathrm{ort}}(\mathbf{z})) = \mathrm{Var}(\widehat{F}_m^{\mathrm{ort}}(\mathbf{z}))$ and $\mathrm{MSE}(\widehat{F}_m^{\mathrm{iid}}(\mathbf{z})) = \mathrm{Var}(\widehat{F}_m^{\mathrm{iid}}(\mathbf{z}))$. We have:

$$\mathrm{Var}(\widehat{F}_m^{\mathrm{iid}}(\mathbf{z})) = \mathbb{E}[(\widehat{F}_m^{\mathrm{iid}}(\mathbf{z}) - \mathbb{E}[\widehat{F}_m^{\mathrm{iid}}(\mathbf{z})])^2] = \mathbb{E}[(\widehat{F}_m^{\mathrm{iid}}(\mathbf{z}))^2] - F^2(\mathbf{z}). \tag{68}$$

Similarly,

$$\mathrm{Var}(\widehat{F}_m^{\mathrm{ort}}(\mathbf{z})) = \mathbb{E}[(\widehat{F}_m^{\mathrm{ort}}(\mathbf{z}))^2] - F^2(\mathbf{z}). \tag{69}$$

We have:

$$\mathbb{E}[(\widehat{F}_m^{\mathrm{iid}}(\mathbf{z}))^2] = \frac{1}{m^2} \sum_{i=1}^{m} \mathbb{E}[(X_i^{\mathrm{iid}})^2] + \frac{1}{m^2} \sum_{i \neq j} \mathbb{E}[X_i^{\mathrm{iid}} X_j^{\mathrm{iid}}]. \tag{70}$$

Similarly, we get:

$$\mathbb{E}[(\widehat{F}_m^{\mathrm{ort}}(\mathbf{z}))^2] = \frac{1}{m^2} \sum_{i=1}^{m} \mathbb{E}[(X_i^{\mathrm{ort}})^2] + \frac{1}{m^2} \sum_{i \neq j} \mathbb{E}[X_i^{\mathrm{ort}} X_j^{\mathrm{ort}}]. \tag{71}$$

Therefore, since marginal distributions of $X_i^{\mathrm{iid}}$ and $X_i^{\mathrm{ort}}$ are the same, we have:

$$\mathrm{MSE}(\widehat{F}_m^{\mathrm{iid}}(\mathbf{z})) - \mathrm{MSE}(\widehat{F}_m^{\mathrm{ort}}(\mathbf{z})) = \binom{m}{2} \cdot 2 \cdot \frac{1}{m^2} (\mathbb{E}[X_1^{\mathrm{iid}} X_2^{\mathrm{iid}}] - \mathbb{E}[X_1^{\mathrm{ort}} X_2^{\mathrm{ort}}])$$
$$= (1 - \frac{1}{m})(\mathbb{E}[X_1^{\mathrm{iid}} X_2^{\mathrm{iid}}] - \mathbb{E}[X_1^{\mathrm{ort}} X_2^{\mathrm{ort}}]) \tag{72}$$

Plugging in the formula for $X_i^{\mathrm{ort}}$ and $X_i^{\mathrm{iid}}$ from Equation 48 and Equation 49, and using our analysis from the proof of Theorem 3 we obtain:

$$\mathrm{MSE}(\widehat{F}_m^{\mathrm{iid}}(\mathbf{z})) - \mathrm{MSE}(\widehat{F}_m^{\mathrm{ort}}(\mathbf{z})) = (1 - \frac{1}{m}) \sum_{t,u=0}^{\infty} a_t a_u \|\mathbf{z}\|_2^{t+u} \mathbb{E}[\|\omega\|_2^t] \mathbb{E}[\|\omega\|_2^u] \cdot$$
$$\frac{\mathbb{E}[r^t]\mathbb{E}[r^u]}{\mathbb{E}[\sqrt{g_1^2 + \dots + g_d^2}^t]\mathbb{E}[\sqrt{g_1^2 + \dots + g_d^2}^u]} (1 - \tau(t, u)). \tag{73}$$

for $\omega \sim \Omega$ and $r \sim \mathcal{N}(0, 1)$. Thus, using Lemma 6, we get:

$$\mathrm{MSE}(\widehat{F}_m^{\mathrm{iid}}(\mathbf{z})) - \mathrm{MSE}(\widehat{F}_m^{\mathrm{ort}}(\mathbf{z})) \geq (1 - \frac{1}{m})\frac{2}{d+2} \sum_{t,u=0}^{\infty} a_t a_u \|\mathbf{z}\|_2^{t+u} \mathbb{E}[\|\omega\|_2^t] \mathbb{E}[\|\omega\|_2^u] \cdot$$
$$\frac{\mathbb{E}[r^t]\mathbb{E}[r^u]}{\mathbb{E}[\sqrt{g_1^2 + \dots + g_d^2}^t]\mathbb{E}[\sqrt{g_1^2 + \dots + g_d^2}^u]} \tag{74}$$
$$= (1 - \frac{1}{m})\frac{2}{d+2} \left( \sum_{t=0}^{\infty} a_t \|\mathbf{z}\|_2^t \mathbb{E}[\|\omega\|_2^t] \cdot \frac{\mathbb{E}[r^t]}{\mathbb{E}[\sqrt{g_1^2 + \dots + g_d^2}^t]} \right)^2 = (1 - \frac{1}{m})\frac{2}{d+2} F_{\Omega,g}^2(\mathbf{z}).$$

That completes the proof. $\qquad\square$

## F.5 PROOF OF THEOREM 4

We showed in the main body of the paper that in contrast to other methods approximating the attention matrix $\mathbf{A}$, our algorithm provides strong concentration guarantees. This is the case also for trigonometric random features, yet, as discussed in the main body of the paper, due to attention renormalization and higher variance of the estimation of small entries of the attention matrix, trigonometric mechanism is sub-optimal. We show here that $m_{\mathrm{opt}}$, the optimal number of random projections for the trigonometric orthogonal mechanism for accurate estimation of the attention matrix does not depend on $L$ but only on $d$. In fact, we prove that if we take $m_{\mathrm{opt}} = \Theta(d \log(d))$, then with $O(Ld^2 \log(d))$-time, we can approximate $\mathbf{A}$ up to any precision, regardless of the number of tokens $L$. In order to provide those guarantees, we leverage recent research on the theory of negative dependence for ORFs (Lin et al., 2020).

We prove the more general version of Theorem 4 from the main body of the paper:

**Theorem 7** (Uniform convergence for the trigonometric mechanism). *Define entries of the attention matrix $\mathbf{A}$ as follows: $\mathbf{A}_{i,j} = g(\mathbf{q}_i^\top) \mathrm{K}(\frac{1}{d^{\frac{1}{4}}} \mathbf{q}_i^\top, \frac{1}{d^{\frac{1}{4}}} \mathbf{k}_j^\top) h(\mathbf{k}_j^\top)$ for some $g, h : \mathbb{R}^d \to \mathbb{R}$ and where $\mathrm{K}$ is a radial basis function (RBF) kernel (Choromanski et al., 2018b) with corresponding spectral distribution $\Omega$ (e.g. Gaussian kernel for which $\Omega = \mathcal{N}(0, \mathbf{I}_d)$). Assume that the rows of matrices $\mathbf{Q}$ and $\mathbf{K}$ are taken from a ball $B(R)$ of radius $R$, centered at $0$ (i.e. norms of queries and keys are upper-bounded by $R$). Define $l = Rd^{-\frac{1}{4}}$ and take $g^* = \max_{\mathbf{x} \in B(l)} |g(\mathbf{x})|$ and $h^* = \max_{\mathbf{x} \in B(l)} |h(\mathbf{x})|$. Then for any $\epsilon > 0$, $\delta = \frac{\epsilon}{g^* h^*}$ and the number of random projections $m = \Omega(\frac{d}{\delta^2} \log(\frac{4\sigma R}{\delta d^{\frac{1}{4}}}))$ for $\sigma = \mathbb{E}_{\omega \sim \Omega}[\omega^\top \omega]$ the following holds: $\|\widehat{\mathbf{A}} - \mathbf{A}\|_\infty \leq \epsilon$ with any constant probability, where $\widehat{\mathbf{A}}$ approximates generalized attention matrix via orthogonal trigonometric random features.*

The result holds in particular for regular softmax-attention for which $\mathrm{K}$ is a Gaussian kernel and $g(\mathbf{x}) = h(\mathbf{x}) = \exp(\frac{\|\mathbf{x}\|^2}{2})$. In that case $m_{\mathrm{opt}} = \Omega(\frac{d}{\delta^2} \log(\frac{4d^{\frac{3}{4}}R}{\delta}))$ since $\sigma = d$.

*Proof.* Let $\mathbf{D_Q}$ be a diagonal matrix with entries of the form: $g(\mathbf{q}_i^\top)$ and let $\mathbf{D_K}$ be a diagonal matrix with entries of the form: $h(\mathbf{k}_i^\top)$. Denote $\mathbf{B} = [\mathrm{K}(\frac{1}{d^{\frac{1}{4}}} \mathbf{q}_i^\top, \frac{1}{d^{\frac{1}{4}}} \mathbf{k}_j^\top)]_{i,j} \in \mathbb{R}^{L \times L}$. Denote by $\widehat{\mathbf{A}}$ and approximation of the attention matrix obtained from trigonometric orthogonal random features and by $\widehat{\mathbf{B}}$ an approximation of matrix $\mathbf{B}$ that those random features provide. We rely on Theorem 3 from (Lin et al., 2020). Note that we can apply it in our case, since for RBF kernels the corresponding functions $f_i$ satisfy $f_1(x) = \sin(x)$, $f_2(x) = \cos(x)$ (thus in particular are bounded). Also, it is not hard to observe (see for instance analysis in Claim 1 from (Rahimi & Recht, 2007)) that we can take: $L_f = 1$ (for $L_f$ as in Theorem 3 from (Lin et al., 2020)). Using Theorem 3 from (Lin et al., 2020), we conclude that:

$$\|\widehat{\mathbf{B}} - \mathbf{B}\|_\infty \leq \delta \tag{75}$$

with any constant probability as long as $m = \Omega(\frac{d}{\delta^2}) \log(\frac{\sigma \cdot \mathrm{diam}(\mathcal{M})}{\delta})$, where $\sigma = \mathbb{E}[\omega^\top \omega]$ and $\mathcal{M}$ is the diameter of the smallest ball $\mathcal{M}$ containing all vectors of the form $\mathbf{z} = \frac{\mathbf{Q}_i}{d^{\frac{1}{4}}} - \frac{\mathbf{K}_j}{d^{\frac{1}{4}}}$. Since $\|\mathbf{Q}_i\|_2, \|\mathbf{K}_j\|_2 \leq R$, we conclude that $\|\mathbf{z}\|_2 \leq \frac{2R}{d^{\frac{1}{4}}}$ and thus one can take $\mathrm{diam}(\mathcal{M}) = \frac{4R}{d^{\frac{1}{4}}}$. We have:

$$\|\widehat{\mathbf{A}} - \mathbf{A}\|_\infty = \|\mathbf{D_Q}(\widehat{\mathbf{B}} - \mathbf{B})\mathbf{D_K}\|_\infty \leq \|\mathbf{D_Q}\|_\infty \|\widehat{\mathbf{B}} - \mathbf{B}\|_\infty \|\mathbf{D_K}\|_\infty \leq \delta g^* h^* \tag{76}$$

Taking $\delta = \frac{\epsilon}{g^* h^*}$ completes the proof. $\qquad \square$

## F.6 DISCUSSION OF THEOREM 4

As a consequence of Theorem 4, the number $m$ of random projections required to approximate the attention matrix within $\epsilon$ error is a function of data dimensionality $d$, the parameter $\epsilon$ and the radius $R$ of the ball within which the queries and keys live:

$$m = \Psi(\epsilon, d, R).$$

The dependence on $d$ and $\epsilon$ is fairly easy to understand: with a larger dimensionality $d$ we need more random projeections (on the order of magnitude $d \log(d)$) to get an approximation within $\epsilon$ error. The dependence on $R$ means that the length of queries and keys cannot grow at a fixed $m$ if we want to retain the quality of the approximation. In particular, this means that FAVOR cannot approximate

hard attention on sequences of unlimited length with a fixed $m$. When the sequence length increases, even the standard attention requires longer and longer vectors to make the softmax concentrated enough to pick single elements. Nevertheless, as seen in our experiments, this limitation does not manifest itself in practice at the lengths we experimented with.

