# OpenReview forum: "Rethinking Attention with Performers"
_ICLR.cc/2021/Conference — ICLR 2021 Oral_

### Official Review · AnonReviewer1 · 2020-10-27
**Great theoretical and experimental results but missing time equalized comparisons with simple baselines**

**Rating:** 7
**Confidence:** 5

**Review:**

### Summary

The authors propose to use the kernel feature map self-attention formulation introduced in [1] to efficiently approximate the softmax attention. The main contribution of the paper lies in the proposed _positive random features_ that can approximate softmax with a strictly positive feature map without which the training is unstable. The authors also show that an approximation of softmax is not necessary for good performance and actually use ReLU random features to achieve their best results when training from scratch.

### Strengths

- The paper deals with a very pressing and important issue, that of the scalability of self-attention.
- The positive random features are also useful outside of the context of self-attention for efficiently approximating softmax.
- The experimental results provide strong evidence about the performance of training transformers with kernel feature-maps.

### Weaknesses

The biggest weakness of the paper in my opinion is the lack of comparison with a simple feature-map as proposed in [1]. Since the authors also use the ReLU random features, we establish that approximating softmax is not necessary for good performance.

1. What would the performance be if a simple deterministic feature map was used?
2. What would it be if the computational cost was equalized either by adding more layers or by increasing the dimensionality of the queries and keys?

The second weakness of the paper concerns the evaluation of the practical softmax approximation capabilities. I find the theoretical results interesting and important but I would like more experimental evidence. Without fine-tuning, the authors provide evidence that the approximation does not work (Fig 5).

1. What would happen, for instance, in a toy task where the Lipschitz constant of the transformer layers was kept low? How big would the feature map need to be in order for the approximation to work in such a simple case?
2. What is being approximated in Fig 4? Is it a randomly initialized attention? What is the rank of the attention matrix?
3. How good would the approximation be for an attention matrix that is almost full rank and how many features would we need then?

### Reasons for my recommendation

I am recommending acceptance because I believe that the positive random features is an important contribution both for transformers and for kernel approximation. In addition, the experimental results are impressive and show that fast kernelized attention indeed works in practice. My only reservation for a higher score is, as mentioned in the weaknesses section, the lack of comparison with simpler feature maps under equalized computation time.

[1]: Transformers are RNNs: Fast Autoregressive Transformers with Linear Attention

---

> ### Author Response · Authors · 2020-11-18
> **Reviewer 1 Rebuttal, Part 1**
>
> We sincerely thank the Reviewer for all the comments.  We address all the points below:
>
> ### **Weakness 1**
>
> #### **1. Simple Deterministic Features**
>
> Thank you for the question! We addressed the question of comparison with Linear Transformers for Reviewer 3, but for Reviewer 1’s convenience, we repeat the answer here.
>
> Experiments conducted by us and other researchers (see: below) comparing Performers with Linear Transformers show that in general the latter are less robust.
>
> For instance, using the attention implementation of the Linear Transformer from https://arxiv.org/abs/2006.16236 resulted in exploding NaN gradients which prevented complete training in the 36-layer ProGen runs for both unidirectional and bidirectional variants (see our comparison curves in the **9th additional rebuttal page** we’ve added to the paper). We did not observe these issues in Performer training.
>
> Furthermore, the comparison with Linear Transformers was actually already independently conducted in the following paper: *Long Range Arena: A Benchmark for Efficient Transformers* (https://arxiv.org/abs/2011.04006). In fact in that paper, Performers are compared against several additional (scalable and not scalable) methods not included in our submission: *Local Attention, Sparse Attention, Longformer, Sinkhorn Transformer, Synthesizer, Big Bird* and the aforementioned *Linear Transformer* on challenging long range context tasks. The results are summarized in Table 1, 2 and Fig. 3. Performers obtain the largest LRA (Long Range Arena) score among all tested **scalable** Transformers methods (which we define by having speed of more than 100 examples/sec), with Linear Transformers being a close competitor (see: Table 1 & Fig. 3). Performers provide the best models for the *Pathfinder-problem* and second best for *Text-problem* and *Image-problem* and get overall LRA score **51.41** across all 5 tasks. In comparison, Linear Transformers provide the best model for the *Text-problem* and get overall LRA score **50.55** across all 5 tasks (see: Table 1). Furthermore, Performers are characterized by the best space/time complexity profiles, as illustrated in Table 2.
>
> **Note:** For reviewers’ convenience, we summarized the main results of that paper (by copy-pasting main tables and figures with the results) on the 9th page of the updated version of our manuscript. Therefore reviewers do not need to explicitly access that paper and can see comparison of different methods with anonymity preserved (i.e. with no information about authors of different algorithms revealed).
>
> Furthermore, in our paper we have already included additional extensive ablation studies with different nonlinearities and kernels using both **deterministic** and random features, see: Appendix, Sec. D3, since the simple deterministic mechanism is a special instantiation of the Performer. Our observations are as follows:
>
> * Nonnegative nonlinear functions for kernels work much better than those that also take negative values.
> * Nonlinear functions for kernels that are not growing too fast are convenient since they do not require additional renormalization to prevent large L2-norm gradients.
> * Positive kernel features are more robust than those taking also negative values (even if the kernel under consideration takes only positive values; this is best demonstrated by comparing the performance of trigonometric and positive random features for the softmax attention approximation).
> * Kernels defined by unbounded nonlinear functions work better than those using bounded ones.
>
> In particular, deterministic features using shifted-eLU/ReLU nonlinearities are reasonable choices, yet we want to emphasize that *approximate softmax with FAVOR+ is the only mechanism providing backward compatibility with regular Transformers* (see: left subfigure of Fig. 5), thus enabling efficient fine-tuning of regular models. In our experiments, approximate softmax with FAVOR+ as well as features with ReLU nonlinearity were the most robust choices across different tasks.

---

> > ### Author Response · Authors · 2020-11-18
> > **Reviewer 1 Rebuttal, Part 2**
> >
> > #### **2. Equalizing computational cost**
> >
> > Thank you very much for the question! We considered many methods of comparing different architectures in a fair way, and ultimately decided that the most sensible one was to fix the computational budget allowed for every run (e.g. every model is allowed to use a 8x8 TPU-v2), as this is one of the most realistic scenarios in practical use cases. This means that the total memory consumption over all accelerators (which is fixed) would be a combination of the model size and batch sizes for a given X-former.
> >
> > We prioritized using the same model sizes for all comparisons (and thus maximized the batch sizes, i.e. sizes of query/key tensors), in particular to show that the Performer’s softmax approximation is very accurate and leads to the same performance as a regular Transformer, as well as to demonstrate that the Performer can outperform other X-formers given the same model sizes.
> >
> > In the example of modeling complexes of protein-sequences, we showed (right subfigure of Fig. 7) that by adding more layers to equalize computational budget, we can train Performer-models substantially outperforming more shallow Transformer variants. We also note that experiments with varying model sizes might in general be difficult to rigorously compare, since increasing model size would imply lowering batch size (within a fixed computational budget) and might have a detrimental effect on the overall performance if the batch size is not large enough. Finally, several solutions for more space efficient training can be actually easily combined together rather than compared with each other. For instance, reversible layers from *Reformer: The Efficient Transformer (2019)* can be easily combined with Performers’ FAVOR+ mechanism.
> >
> > ### **Weakness 2**
> >
> > * **Lipschitz-continuity:** Thank you for the good question. Formally speaking, Transformers are not Lipschitz-continuous because they incorporate layer normalization, where the hidden state is divided by the standard error of its entries. Hence, when the standard error is small enough (i.e. entries of the hidden state are almost equal), the gradient through layernorm can be of an arbitrary magnitude. Layer normalization is crucial for empirically stable training and is used in all Transformer architectures we are aware of. Therefore, even in simple setups, we cannot guarantee a small Lipschitz constant.
> >
> > * **Fig. 4:** To clarify our setup, we pass in the same randomly generated (but properly scaled) $Q, K, V$ matrices into both the exact softmax attention, as well as our tested approximation mechanisms, and compute their empirical mean squared errors. We reran the experiment and checked that all generated ground-truth attention matrices $(A)$ are full-rank. See also our response below regarding the rank of the attention matrix $A$.
> >
> > * **Approximation of the full-rank attention matrices $A$:** Thank you very much for a good question. We also addressed it in the response to Reviewer 2 (see: *“Low/Full-Rankness”* section), where in particular we explained the difference between the common SVD approximation and FAVOR’s approach.
> >
> > We would like to clarify that our presented upper bounds for the number of random features m needed for accurate estimation of the attention matrix via FAVOR is completely independent of the rank of $A$. Therefore our bounds are useful in practical applications, where we cannot assume that $A$ is low-rank. We do not exclude the possibility that extra structural assumptions regarding $A$ (such as low-rankness) might lead to even stronger upper bounds on $m$, yet we do not rely on any such assumptions. We would like to explore this topic in more detail in future work.

---

### Official Review · AnonReviewer3 · 2020-10-28
**Theoretically grounded O(N) approximation of the softmax attention**

**Rating:** 8
**Confidence:** 4

**Review:**

##########################################################################

Summary:


The paper proposed a theoretically grounded O(N) approximation of the softmax attention. The key idea is to interpret attention as a kernel function and construct the random feature projection that can reproduce this kernel. It is highly non-trivial to derive a feature mapping that can accurately approximate the softmax kernel. To better approximate the softmax kernel, the author proposed some important design choices, all of which are supported by theoretical and empirical evidences. The author showed that 1) adopting non-negative random features is very essential to the approximation and the proposed Positive Random Features (PRF) can effectively reduce the variance when the attention values are small, 2) drawing orthogonal random matrices can further reduce the variance of the approximation, 3) the final proposed Performer model runs faster, takes less memory, and has better performance than other O(N) and O(N logN) attention methods.


##########################################################################

Reasons for score:


The paper is very well-written and should be accepted. This is an important landmark in the research about O(N) attention. The design of the random feature mapping is reasonable and theoretical analysis is convincing. Experiments show that Performer is better than the other O(N) attention methods and also other efficient attention methods.


##########################################################################

Pros:


1. The paper gives a provable O(N) approximation of the softmax attention. The method works without assumptions on the structure of the attention map (like sparsity). The theoretical proofs provide good insights on how to design a good O(N)-complexity approximation to the  attention mechanism.

2. Apart from approximating the softmax attention, the proposed FAVOR+ method can be utilized to approximate other attention kernels. In fact, the author has experimented with Performer-ReLU, which outperforms Performer in some experiments. This provides the insight that softmax attention may not be the best choice.

3. The author conducted very comprehensive ablation studies on different components of the proposed method. This includes: 1) effectiveness of using the positive features, 2) drawing orthogonal random samples, 3) redrawing the random samples


##########################################################################

Cons:


1. From Figure 5 and also Figure 15, periodic redrawing is quite essential. However, the author has not mentioned about the implementation details on how they redraw the random samples and how to choose the period. For me, I feel that this hyper-parameter should be important because the model may need to ensure that each group of samples has been trained for a sufficient amount of time.

2. Performer-ReLU has replaced the attention kernel and can sometimes be better than the softmax attention. Thus, I feel the author may also want to compare with the linear attention method in ((Katharopoulos et al., 2020) "Transformers are RNNs: Fast Autoregressive Transformers with Linear Attention".


##########################################################################

Questions during rebuttal period:


Please address and clarify the cons above


#########################################################################

Typos:

(1) Page 5, after "than those from SM_{2m}(x, y)", there is an additional right bracket.

---

> ### Author Response · Authors · 2020-11-18
> **Reviewer 3 Rebuttal, Part 1**
>
>
> We sincerely thank the Reviewer for all the comments.  We address all the points below.
>
> ### **Redrawing**
>
> Thank you for a very good question. In all results involving redrawing, by default we redraw features (via a new random seed) after every gradient step.
>
> Currently, for the 36 Layer TrEMBL experiment (where $L = 1024$), redrawing leads to 0.8 train steps/second, while no-redrawing leads to 1.6 train steps/second. For our PG-19 experiment ($L=1024$), no-redraw leads to 2.4 train steps/second, while redraw leads to 1.7 train steps/second. However, redrawing becomes negligible for longer sequence lengths, as both redraw and no-redraw variants for a 6-layer regular model on ImageNet64 ($L = 12288$) have 0.2 train steps/second, due to the rest of the pipeline dominating computational costs.
>
> The original redrawing procedure reduces to conducting Gram-Schmidt orthogonalization on $d \times d$ blocks, followed by the renormalization of each row of the resulting matrix via $\text{chi}(d)$-distribution, so that marginal distributions remain Gaussian (we will clarify this in the final version of the paper). Even this can be further improved by applying as random samples rows of: [1] random Hadamard matrices or [2] products of Givens random rotations (see: Appendix, Sec. B.2). These in practice provide accurate proxies of Gaussian orthogonal ensembles, yet do not require any orthogonalization (since orthogonality is naturally embedded in the overall mechanism via Hadamard/Given rotation-matrices that are orthogonal) and effectively make redrawing cost negligible even for moderate $L$ regimes.
>
> Note that one can also apply an adaptive redrawing schedule. We observed that redrawing is mainly important in the initial stages of training, but its frequency can be lowered in the later stages, when it is less useful. Such a strategy can be defined by a dataset-dependent hyperparameter.

---

> > ### Author Response · Authors · 2020-11-18
> > **Reviewer 3 Rebuttal, Part 2**
> >
> > ### **Comparison with Linear Attention Method**
> >
> > Thank you for the question!
> >
> > The attention mechanism of Linear Transformers (that uses shifted elu nonlinearity) is a special instantiation of the generalized attention from Performers. In fact we already run extensive ablation studies over kernels defined by different nonlinearities in the Appendix, Sec. D3. Experiments conducted by us and independently by other researchers (see: below) comparing Performers with Linear Transformers show that in general the latter are less robust.
> >
> > For instance, using the attention implementation of the Linear Transformer from https://arxiv.org/abs/2006.16236 resulted in exploding NaN gradients which prevented complete training in the 36-layer ProGen runs for both unidirectional and bidirectional variants (see our comparison curves in the **9th additional rebuttal page** we’ve added to the paper). We did not observe these issues in Performer training.
> >
> > Furthermore, the comparison with Linear Transformers was actually already independently conducted in the following paper: *Long Range Arena: A Benchmark for Efficient Transformers* (https://arxiv.org/abs/2011.04006). In fact in that paper, Performers are compared against several additional (scalable and not scalable) methods not included in our submission: *Local Attention, Sparse Attention, Longformer, Sinkhorn Transformer, Synthesizer, Big Bird* and the aforementioned *Linear Transformer* on challenging long range context tasks. The results are summarized in Table 1, 2 and Fig. 3. Performers obtain the largest LRA (Long Range Arena) score among all tested **scalable** Transformers methods (which we define by having speed of more than 100 examples/sec), with Linear Transformers being a close competitor (see: Table 1 & Fig. 3). Performers provide the best models for the *Pathfinder-problem* and second best for *Text-problem* and *Image-problem* and get overall LRA score **51.41** across all 5 tasks. In comparison, Linear Transformers provide the best model for the *Text-problem* and get overall LRA score **50.55** across all 5 tasks (see: Table 1). Furthermore, Performers are characterized by the best space/time complexity profiles, as illustrated in Table 2.
> >
> > **Note:** For reviewers’ convenience, we summarized the main results of that paper (by copy-pasting main tables and figures with the results) on the 9th page of the updated version of our manuscript. Therefore reviewers do not need to explicitly access that paper and can see comparison of different methods with anonymity preserved (i.e. with no information about authors of different algorithms revealed).
> >
> > **Typos:** We fixed all these typos in the updated version of the paper.

---

### Official Review · AnonReviewer2 · 2020-10-28
**A solid paper that will likely affect the way Transformers are estimated**

**Rating:** 8
**Confidence:** 4

**Review:**

This is a solid paper that presents a computationally less expensive, unbiased, low-variance estimator of the Transformer architecture.

**Strengths**:
1. The authors provide mathematical guarantees for the suggested estimator.
2. The estimator (called FAVOR+) seems a more scalable replacement for regular attention.
3. Theory is empirically verified in a variety of experimental settings.

**Question**:
In Introduction, you mention that your method does not rely on low-rankness, but doesn't your equation (4) as well as Fig. 1 assume that the attention matrix $\mathbf{A}$ allows low-rank decomposition? (I suppose $r<d$)

**Suggestion**:
Will it be possible to compare a pre-trained Performer-based encoder (like BERT) against its competitors (Linformer, Reformer) on a [GLUE benchmark](https://openreview.net/pdf?id=rJ4km2R5t7) or on [probing tasks](https://openreview.net/forum?id=SJzSgnRcKX)? I see that you compare them (Per-/Lin-/Re-former) against the original Transformer on the pre-training task itself (Fig. 5), but I am wondering how well the pre-training performance transfers to downstream tasks in each case.

**Minor issues**:
1. In the proof of Lemma 1 you first prove that $\exp(\left\lVert\mathbf{x}+\mathbf{y}\right\rVert^2/2)=\mathbb{E}_{\omega\sim\mathcal{N}(\mathbf{0},\mathbf{I})}[\exp(\omega^\top\mathbf{x})\cdot\exp(\omega^\top\mathbf{y})]$, however this is simply an m.g.f. of $\omega$ at $\mathbf{x}+\mathbf{y}$, and thus IMHO does not need to be redirived. (I assume that the formula for the m.g.f. of a multivariate Gaussian random vector is well-known).
2. Similarly, eq. (16) is the m.g.f. of $\omega\sim\mathcal{N}(\mathbf{0},\mathbf{I})$, it does not have to "immediately follow from" some other fact.

**Typos**:
1. In eq. (18), a norm is missing in the last term: $\mathbf{z}^2$ -> $\left\lVert\mathbf{z}\right\rVert^2$.
2. In the text after eq. (18): inequality -> equality

**Update after the author's response**: The authors have answered my questions during the rebuttal period and I am satisfied with the response. Hence updating my score: 7 $\to$ 8.

---

> ### Author Response · Authors · 2020-11-18
> **Reviewer 2 Rebuttal, Part 1**
>
> We sincerely thank the Reviewer for all the comments. We address all the points below.
>
> ### **Low/Full-Rankness**
>
> What we meant is that we do not need to assume that the original attention matrix $A$ is low-rank since we can accurately and simultaneously estimate **all entries** of $A$ (even if it is full-rank) using our estimator. However as the Reviewer noticed, since the estimator is a product of two lower rank matrices, the approximating matrix is also lower rank. So effectively in our setting, a lower rank matrix is accurately and **unbiasedly** estimating a potentially full-rank matrix. We will clarify this in the final version of the paper.
>
> As an analogy, a full-rank matrix $A$ can still be approximated by taking some of the singular vectors of the SVD decomposition of $A$. The difference between the SVD and our method, is that the former is a deterministic (and thus biased) approximation applicable to any real-valued matrix, while FAVOR+ is a completely different unbiased randomized approximation exploiting the structure of the set of attention matrices that can be expressed via kernels as: $A_{i,j}=K(q_{i},k_{j})$, e.g. $A_{i,j} = \exp (q_{i} * k_{j}^{T})$ (as in the softmax case).
>
> For the SVD case, since the matrix can be arbitrary, the error of SVD-based approximation (via the first $k$-significant vectors) can be arbitrarily high (e.g. by simply making the singular values equal).  This is not the case for FAVOR+, which provides strong theoretical upper bounds on the error with the number of random features $m$ **independent** of the rank of $A$. Our upper bounds on $m$ are not tight and thus in principle it might be the case that additional structural assumptions regarding $A$, such as low-rankness, can lead to even fewer features $m$ needed by FAVOR+, yet we want to emphasize that we do not need such additional assumptions.

---

> > ### Author Response · Authors · 2020-11-18
> > **Reviewer 2 Rebuttal, Part 2**
> >
> >
> > ### **Downstream Tasks - suggestions**
> >
> > Thank you very much for the comment! We will add additional results comparing different architectures on the downstream tasks in the final version of the paper. In fact several downstream-tasks experiments were already conducted in other papers. For instance, in the following paper: Long Range Arena: A Benchmark for Efficient Transformers (https://arxiv.org/abs/2011.04006), Performers are compared against several additional (scalable and not scalable) methods not included in our submission: *Local Attention, Sparse Attention, Longformer, Sinkhorn Transformer, Synthesizer, Big Bird* and the aforementioned *Linear Transformer* on challenging long range context tasks. Performers obtain the largest LRA (Long Range Arena) score among all tested **scalable** Transformers methods (which we define by having speed of more than 100 examples/sec).
> >
> > **Note:** For reviewers’ convenience, we summarized the main results of that paper (by copy-pasting main tables and figures with the results) on the **9th page of the updated version of our manuscript.** Therefore reviewers do not need to explicitly access that paper and can see a comparison of different methods with anonymity preserved (i.e. with no information about authors of different algorithms revealed).
> >
> > Tested tasks from that paper involved in particular [we included the authors’ description]:
> >
> > * **A longer variation of the standard ListOps task proposed in [Nangia & Bowman, 2018],** which was designed to investigate the parsing ability of neural models.
> >
> > * **Byte-level text classification using real-world data.** “This task also benchmarks the ability of the models to deal with compositionality as it is required to compose characters into words into higher-level phrases. Compared to ListOps, boundaries are less well defined and need to be learned from the data, which is a challenging problem in its own right (Kawakami et al., 2019).”
> >
> > * **Byte-level document retrieval.** “This task is mainly about modeling a similarity score between two documents in a ‘two tower setup’ in which compressed representations are concatenated and passed into a linear classifier. Note that we deliberately prevent models from using cross attention. This task thus serves as a test of how well models are able to compress long sequences into representations suitable for similarity-based matching. We use the ACL Anthology Network (AAN; Radev et al., 2013) dataset, which identifies if two papers have a citation link, a common setup used in long-form document matching (Jiang et al., 2019; Yang et al., 2020).”
> >
> > * **Image classification on sequences of pixels.**  CIFAR-10 dataset (Krizhevsky, 2009) was used for the image classification task.
> >
> > * **Pathfinder (long-range spatial dependency).** “The Pathfinder challenge (Linsley et al., 2018; Kim* et al., 2020) was first introduced for learning long-range spatial dependencies. It is a synthetic visual task motivated by cognitive psychology (Houtkamp & Roelfsema, 2010). The task requires a model to make a binary decision whether two points represented as circles are connected by a path consisting of dashes.”
> >
> > **Theory:** Thank you very much for all the suggestions. They will be incorporated in the final version of the paper.
> >
> > **Typos:** We fixed all these typos in the updated version of the paper.

---

> > > ### Comment · AnonReviewer2 · 2020-11-19
> > > **I am almost satisfied with the author response**
> > >
> > > Thank you for clarifications on low/full-rankness, and for pointing to the LRA benchmark! I am overall satisfied with your response.
> > >
> > > I am glad to see that the Performer does an excellent job on LRA. However, there are two issues with this benchmark:
> > > - fine-tuning/probing pre-trained models is not the scope of the LRA
> > > - many tasks seem unorthodox to me (but this may be my biased opinion)
> > >
> > > I would like to see the following: both vanilla Transformer and Performer are first pre-trained on a massive amount of data (probably in a self-supervised fashion) and then both are either fine-tuned or probed using a small amount of annotated data. If you think that this is too much for the rebuttal period, I am totally ok with the existing evaluation.

---

> > > > ### Author Response · Authors · 2020-11-23
> > > > **Additional Experiments**
> > > >
> > > > We would like to sincerely thank the Reviewer for the comment.  It would be definitely insightful to run the proposed experiments. Given the time range of the rebuttal period, we leave to future work running comprehensive additional studies regarding pre-training and fine-tuning / probing.
> > > >
> > > > We want to mention that in addition to LRA tasks, Performers were also recently independently applied for end-to-end speech recognition in Conformers (with Performer's attention module replacing the regular one), and the results are reported in the paper: *"Efficient End-to-End Speech Recognition Using Performers in Conformers"* (https://arxiv.org/abs/2011.04196v1). Again, as before, to preserve anonymity, we summarize the results below so that Reviewer does not need to read that paper:
> > > >
> > > > **Results:** Performers yield competitive performance on the LibriSpeech corpus (the training set contains 960 hours of read English speech) with **10 million** parameters and only linear computational complexity. For comparison, the regular Conformer model consists of **116.4 million** parameters. The proposed Performer-based architecture also outperforms the dynamic and lightweight convolution approach (Y. Fujita, A. S. Subramanian, M. Omachi, and S. Watanabe, “Attention-based ASR with Lightweight and Dynamic Convolutions,” in Proc. of ICASSP. IEEE, 2020, pp. 7034–7038.) by about **20%** relatively in word error rate, with a substantially smaller model size.

---

> > > > > ### Comment · AnonReviewer2 · 2020-11-24
> > > > > **Score update**
> > > > >
> > > > > Ok, I see that the performer has already started to be used in applied tasks. I think we will soon see a Performer-based BERT.
> > > > >
> > > > > I believe that your submission should be in the top 50% of accepted papers, so I'm changing my grade from 7 to 8.

---

### Official Review · AnonReviewer4 · 2020-10-29
**Another step forward towards efficient transformers**

**Rating:** 7
**Confidence:** 3

**Review:**

This paper proposes a set of techniques called Fast Attention Via positive Orthogonal Random features (FAVOR+) to approximate softmax self attention in Transformers and achieve better space and time complexity when the sequence length is much higher than feature dimensions ($L\gg d$). The resulting architecture, Performers, is provably and practically accurate in estimating regular full-rank attention without relying on any priors such as sparsity or low-rankness. It can also be applied to efficiently model other kernalizable attention mechanisms beyond softmax, achieving better empirical results than regular Transformers on some datasets with such strong representation power. Performers are tested on a rich set of tasks including pixel-prediction, language modeling and protein sequence modeling, and demonstrated competitive results with other examined efficient sparse and dense attention models. I think this paper is a solid step towards more efficient Transformers for practical long-sequence data, and should be accepted. Below I raise two potential questions and look forward to the solutions in the future.

1. For Performers, speedup can only be achieved when $L\gg d$, which means Performers might have less advantages on wider Transformers. The time complexity of Performers for approximating the attention matrix is $O(Ld^2\log(d))$, while a regular Transformer has a time complexity of $O(L^2d)$ to compute the attention matrix. From Figure 3, it looks like for a model similar to BERT-Base, Performers is only faster when the sequence length is larger than 512. I wonder how wide Performers can be while  preserving faster speed than Transformers with the same size on the current datasets, and how their test scores compare.

2. It looks like Performers do not converge as fast as Transformers on larger language modeling datasets like PG-19. I wonder whether similar phenomenon will happen in other domains when the dataset is larger, and feel quite curious about the cause.

---

> ### Author Response · Authors · 2020-11-18
> **Reviewer 4 Rebuttal**
>
> We sincerely thank the Reviewer for all the comments. Below we address the two questions raised:
>
> ### **Question 1**
>
> Thank you for an excellent question.
>
> Time complexity of FAVOR+, given sequence length $L$, number of kernel features $m$, and embeddings’ dimension $d$, is $O(Lmd)$, and the optimal choice of $m$ reflects the tradeoff between approximation accuracy and speed.
>
> In order for FAVOR+ to strictly beat the $O(L^{2} d)$ runtime of the regular Transformer, we only need $L \gg m$ (i.e. the breakpoint on $L$ is only dependent on $m$). This is also true regarding space complexity.
>
> Using $m = d  \log(d)$ features suffices to get a *provably* accurate approximation of the attention matrix, which in turn leads to the $O(L d^{2} \log(d))$ runtime. However, we noticed that *in practice*, one can often use $m \ll d  \log(d)$ without noticeable loss of accuracy (we by default took $m=256$ for different $d$ and noticed that increasing $m$ further does not impact the results). This means that the Performer may still provide speedups or at least does not suffer from slowdowns, even when $L < d$.
>
> We further empirically verified, by using a regular 6-layer model at $(\text{batch size}=1, L=1024, d=4096, m=256$), that the Performer works well also when $L < d$ (large width setting):
>
> 1. **(Memory)** Both Performer and Transformer achieved their maximum possible power-of-2 size embedding dimensionality of $d=2^{12}=4096$ before running out of GPU memory.
>
> 2. **(Runtime)** Forward pass speeds were (Performer = 0.08 sec, Transformer = 0.08 sec) and backward pass speeds were (Performer =  0.29 sec, Transformer = 0.31 sec).
>
> We want to emphasize that Performers are also compatible with general attention (using random or deterministic features), where often $m \ll d \log(d)$. Detailed analysis of large-width Transformers through the lens of efficient scalable methods is undoubtedly an important future research area.
>
> ### **Question 2**
>
> Performers’ results for PG-19 in the main body of the paper are for the approximate softmax version, and slower convergence with sudden loss drops might be partially caused by variance coming from random features (we have already demonstrated that in particular, if random features are not periodically redrawn, training can converge to sub-optimal models). This sudden loss drop phenomenon occurs also when training Performer-Softmax on ImageNet64, although in this case we do not know the result of the regular Transformer, which simply runs out of memory. However, we noticed that for the TrEMBL bioinformatics dataset, **(Fig. 6),** convergence profiles of regular Transformers and Performers are smoother and more similar (after a very initial phase of training).
>
> Note that in terms of relative dataset sizes, PG-19 is the smallest, next is ImageNet64, and TrEMBL is the largest, but the Performer converges more smoothly on TrEMBL than PG-19. These results suggest that the relative rate of convergence heavily depends on the intrinsic features of the dataset, rather than the size of the dataset. We plan to conduct more detailed analysis of this phenomenon in future work, in particular to understand what dataset features are most important and thus reveal which datasets these approaches will be most effective for.
>
> Regardless, the main goal of the PG-19 experiments was to show that approximate softmax via FAVOR tightly estimates regular softmax-attention and outperforms other methods, so that in the setting where running regular softmax Transformers is not feasible (e.g. ImageNet64), approximate softmax becomes its natural replacement.

---

### Comment · ~Jin-Hwa_Kim1 · 2021-05-25
**Questions on the proof of Theorem 5**

I really enjoyed reading this paper with a nice idea on the approximation of softmax-kernel. Unfortunately, after proofreading the Appendix I encounter some issues I believe. Could you clarify these?

1. How do you come with Eqn. 56? I assume that you used the same set of canonical basis vectors $e_1, ..., e_m$, but use their independence in the i.i.d. samples. However, I believe, to derive Equation 56, we should use independent random unit vectors instead of $e_1, ..., e_m$ in Eqn.53 since $g$ is fixed. In that case, $g_i^{d_i}$ should be the linear combination of {$g_i$} to $d_i$.

2. I assume that Eqn. 61 is a re-written form of Eqn. 56. However, you regard here $g_i^{d_i}$ and $\sqrt{g_1^2+\dots+g_d^2}^{d_i}$ are independent, but it's not. Therefore we cannot have Eqn. 61 from Eqn. 56.

Typos in F.4.2. Proof of Theorem 5:
1) After Eqn. 49, "int" the formula -> "in" the formula
2) In Eqn. 50, there is a missing coefficient term for $\hat{\Delta}$.
3) After Eqn. 63, "firsr" -> "first",
$\Delta(\hat{(d_1,\dots,d_m)})$ -> $\hat{\Delta}(d_1,\dots,d_m)$,
"at least of of" -> "at least one of"
4) a missing expectation in the LHS of Eqn. 65 and the missing coefficient for L like in Eqn. 50.

---

> ### Comment · ~Krzysztof_Marcin_Choromanski1 · 2021-05-25
> **Answering questions regarding proof of Theorem 5**
>
> Thank you very much for all the comments. We are happy you enjoyed the paper. Let us answer your questions below.
>
> 1. The first equality in Eq. (56) comes from the independence of omega^{iid}_{i}. To get the second equality we observe that: directions of omegas are chosen independently from their lengths, and furthermore omega is isotropic. Thus the distribution of the dot-product of the L2-normalized omega_i with any fixed deterministic unit-length vector is exactly the same as a distribution of
> g_i / \sqrt{g_{1}^{2}+...+g_{d}^{2}}, where g is sampled from N(0,I_d).
>
> 2. I think there is some misunderstanding here. Those two expressions are indeed not independent, but the beautiful thing about Gaussian distribution is that the equality still holds and we *do not need independence*. The reason for that is Lemma 5. At first glance when you look at it, you might think that Eq. (57) requires independence of some of the expressions involved in the formula but they key observation is that it does not. The proof is actually very simple and is just few-lines long.
>
> Thank you for catching the typos ! And thank you for interesting questions regarding the proof.

---

> > ### Comment · ~Jin-Hwa_Kim1 · 2021-05-27
> > **Thanks for the reply! Comment 2 raises an inquiry.**
> >
> > 1. The "any fixed deterministic unit-length" makes sense to me.
> > 2. I checked with Lemma 5. If the statement of $r$ is correct, we can get:
> >
> > $E[r_1^{k_1}] \cdot \dots \cdot E[r_1^{k_s}]$ = $\Pi_{i=1}^{s}$ $E[ \frac{g_{1}^{k_i}} { ❘❘\mathrm{g}❘❘_{2}^{k_i}} ]$ $\Pi_{i=1}^{s} E[ ❘❘\tilde{\mathrm{g}}❘❘_2^{k_i} ]$ (a simpler arrangement than in Eqn. 58)
> >
> > This states Eqn. 61 comes from Eqn. 56.
> >
> > However, the related question is raised, is it fact that:
> >
> >  $\mathrm{r} \sim \mathrm{g}$ where $\mathrm{r} = \frac{\mathrm{g}}{❘❘\mathrm{g}❘❘_2}❘❘\mathrm{\tilde{g}}❘❘_2$.
> >
> > in the proof of Lemma 5? It seems to state the distribution of $\mathrm{g}$ is not affected by the change of the magnitude ratio of $❘❘\mathrm{g}❘❘_{2}$ and an independent copy of it, but I have no idea how to proof this simple statement considering the dependency of $\mathrm{g}$ and $\mathrm{❘❘g❘❘_2}$. Could you elaborate this statement kindly?

---

> > > ### Comment · ~Krzysztof_Marcin_Choromanski1 · 2021-05-27
> > > **Additional clarification**
> > >
> > > Let me elaborate on r ~ g. Note that by the definition of r we have:
> > >
> > > 1. r is isotropic (since g is isotropic) with direction chosen independently from its magnitude
> > > 2. the distribution of the length of r is exactly xi(d) (since length (g / length(g))=1)
> > >
> > > But this is the definition of the multivariate Gaussian distribution with covariance matrix I_d. So indeed:
> > > r ~ g.

---

> > > > ### Comment · ~Jin-Hwa_Kim1 · 2021-05-27
> > > > **I really appreciate your kind explanation.**
> > > >
> > > > I believe your explanation makes sense. Thank you very much!

---

> > > > > ### Comment · ~Krzysztof_Marcin_Choromanski1 · 2021-05-27
> > > > > **clarification**
> > > > >
> > > > > Thank you very much for the questions :) We are happy we could help.

---

### Comment · ~Steve_Siu1 · 2021-07-04
**Question on the proof of Theorem 1**

Thanks for writing this very interesting paper,  I am learning alot from it. I am currently going through the proof of Theorem 1, and there's a step I don't understand. Could you please clarify it?

In particular, I don't know how to derive equation (21). In the infinite sum, instead of getting the 2-norm $||z||_{2}^{2k}$ for all $k$, I am getting the $2k$-norm $||z||^{2k}\_{2k}$. Could you please give us a sketch of the proof of equation (21)?

Here's my derivation for equation (21): https://imgur.com/a/6BNSsbh

Thanks!

---

### Comment · ~Bunlong_Lay1 · 2021-08-30
**A few questions**

Nice and interesting paper. However, I have a few questions.
1) In regard of Lemma 2, you compute the MSE of the positive SM. Then you deduce that if SM tends to 0, then also the MSE of the positive SM. This is correct, but what happens if $SM(x,y) \neq 0$ and $x,y$ becomes big. In that case the MSE explodes as well ? Why isn't this a problem?
2) Section 4.2 in Figure 4 you wrote d=16. Is this a typo? What feature dimension did you really take? Also, which estimators are shown in the left plot (I assume it is cos/sin) ?
3) A more general question. I found it unintuitive, that the you take only $m<d$ feature samples to approximate the expectation in equation 3. The expectation (or integral) for example of the positive SM is with respect to $\omega$ that it is taken from $\mathbb{R}^d$. So now taking only a few sampling points $\omega_i$ as in equation (5) is not a sufficient approximation of an integral over  $\mathbb{R}^d$. What guarantees that this approximation is sufficient?

---

### Decision · Program_Chairs · 2021-01-07
**Final Decision**

**Decision:**

Accept (Oral)

**Comment:**

This is a solid paper that proposes a new method for approximating softmax attention in transformer architectures that scales linearly with the size of the sequence. Even though linear architectures have been proposed before using a similar idea (Katharopoulos et al 2020), this paper provides a better solution along with theoretical analysis and makes a rigorous empirical comparison against other methods. All reviewers agree that this is a strong paper that should be accepted. I suggest citing the recent paper https://arxiv.org/abs/2011.04006 (Long Range Arena, mentioned in the discussion) which provides further comparisons on long-range benchmarks, including the method presented in this paper and Katharopoulos et al 2020, along with a detailed discussion of the differences between the two methods.